# Brain-wide mapping of neural activity controlling zebrafish exploratory locomotion

Timothy W Dunn[1,2,3†], Yu Mu[3†], Sujatha Narayan[3], Owen Randlett[1],
Eva A Naumann[1,4], Chao-Tsung Yang[3], Alexander F Schier[1,2], Jeremy Freeman[3‡],
Florian Engert[1,2‡], Misha B Ahrens[3*‡]

[1]Department of Molecular and Cellular Biology, Harvard University, Cambridge,
United States; [2]Program in Neuroscience, Department of Neurobiology, Harvard
Medical School, Boston, United States; [3]Janelia Research Campus, Howard Hughes
Medical Institute, Ashburn, United States; [4]Department of Neuroscience, Physiology
and Pharmacology, University College London, London, United Kingdom

**Abstract** In the absence of salient sensory cues to guide behavior, animals must still execute
sequences of motor actions in order to forage and explore. How such successive motor actions are
coordinated to form global locomotion trajectories is unknown. We mapped the structure of larval
zebrafish swim trajectories in homogeneous environments and found that trajectories were
characterized by alternating sequences of repeated turns to the left and to the right. Using whole-
brain light-sheet imaging, we identified activity relating to the behavior in specific neural
populations that we termed the anterior rhombencephalic turning region (ARTR). ARTR
perturbations biased swim direction and reduced the dependence of turn direction on turn history,
indicating that the ARTR is part of a network generating the temporal correlations in turn direction.
We also find suggestive evidence for ARTR mutual inhibition and ARTR projections to premotor
neurons. Finally, simulations suggest the observed turn sequences may underlie efficient
exploration of local environments.

*For correspondence: ahrensm@
janelia.hhmi.org

†These authors contributed
equally to this work
‡These authors also contributed
equally to this work

**Competing interests:** The
authors declare that no
competing interests exist.

**Reviewing editor:** Ronald L
Calabrese, Emory University,
United States

## Introduction

Locomotion trajectories must be highly structured to achieve goals that cannot be reached by indi-
vidual motor actions (*Flavell et al., 2013*; *Berman et al., 2014*; *Maye et al., 2007*). When environ-
mental cues such as odor gradients or visual landmarks are present, these can be used to guide
sequential motor actions in a goal-driven manner, such as navigating up an odor gradient toward
food (*Gomez-Marin et al., 2010*). When such cues are lacking, however, behavior must be struc-
tured so that efficient foraging and exploration can continue until cues are found. This behavior
often follows optimal algorithms (*Stephens, 1986*; *Charnov, 1976*) that must be guided by internal
brain activity. Such internal activity is necessarily embedded in brain-wide patterns of spontaneous
activity, but the relevant signals and brain areas remain elusive.

Decades of motor system research have identified many regions involved in the initiation of loco-
motion. The basal ganglia, which are closely associated with action selection, have been studied
extensively in mammals and more recently in lamprey (*Stephenson-Jones et al., 2011*), and the
mesencephalic locomotor region (MLR) (*Sirota et al., 2000*; *Dubuc et al., 2008*) and the dience-
phalic locomotor region (DLR) (*El Manira et al., 1997*) are causally linked to coordinated motor out-
put. However, many other regions show activity related to motor patterns (e.g. [*Arrenberg et al.,
2009*]), and it is possible that many adjunct motor centers have yet to be characterized. In any case,

**eLife digest** Much of an animal's behavior is guided by cues in the environment: many animals follow odors to find food, for example. But even in the absence of such cues, animals continue to show spontaneous behaviors that are optimized to help them discover resources, such as food, or landmarks, such as shelter. While these behaviors have been observed in many animals, it is unclear how they are supported by the nervous system. This is partly because it is hard to know where to look for relevant signals in large brains of many animals.

The development of whole-brain imaging techniques in zebrafish larvae offers a possible solution to this problem. Zebrafish are commonly used in laboratory studies because the zebrafish genome has been fully sequenced and they reproduce quickly. Whole-brain imaging in larval zebrafish has previously revealed widespread and complex patterns of spontaneous activity. However, it has been unclear whether or how these 'thoughts' are translated into behavior. Moreover, while researchers have studied how the fish respond to lights and sounds, little is known about how fish behave in the absence of guiding stimuli from their environment.

Dunn, Mu et al. now show that spontaneous fish behavior is not random, but is instead characterized by alternating states in which the fish are more likely to repeatedly turn either left or right. Simulations show that this pattern of swimming increases the fish's local foraging efficiency. By analyzing data from across the whole brain, Dunn, Mu et al. identified specific circuits of neurons that help generate these switching chains of turns. This alternating left-right rhythm appears to be dictated by signals sent between theses sets of neurons and may be supported by feedback from the behavior itself.

This analysis generates specific predictions about how specific neurons should connect with one another, and about the relationship between this connectivity and the activity of the rest of the brain. Future studies are required to test these predictions, and to determine how factors – such as whether an animal is hungry, for example – influence the pattern of spontaneous movements.

it is likely that multiple circuits participate in shaping the fine structure of motor output. Furthermore, it is unknown how higher level structure in spontaneous behavior, potentially reflecting goal-driven internal states like those underlying foraging and surveillance, might be realized in circuitry across the vertebrate brain.

We investigated the spatiotemporal properties of internally generated actions by characterizing the spontaneous locomotion patterns of larval zebrafish swimming in an equiluminant environment devoid of explicit sensory cues. Even without structured external cues, fish remain highly active, swimming and turning in discrete bouts. While this behavior appears random, analysis of turn sequences over time revealed a specific temporal structure: a turn is likely to follow in the same direction as the preceding turn, creating alternating 'chains' of turns biased to one side. Overall, such a pattern generates conspicuous, slaloming swim trajectories. These are distinct from biased random walks, which evolve randomly at any point in time, and instead show a strong dependency on past behavior. Because the timescales of spatiotemporal correlations exceeded the timescales of individual swim bouts, we hypothesized that networks upstream of the peripheral motor system likely govern this unique pattern of spontaneous behavior.

Given the large space of putative brain circuits underlying this behavioral program, we employed novel techniques to search for neural populations controlling spontaneous turning. Whole-brain imaging during behavior is a promising method for finding unknown neuronal populations (*Ahrens et al., 2013*; *2012*; *Portugues et al., 2014*; *Vladimirov et al., 2014*), which, in contrast to conventional recordings from subsets of brain areas, increases the likelihood that neurons underlying a specific behavior will be discovered. To this end, we combined a fictive version of the spontaneous behavior with light-sheet imaging, enabling fast, volumetric whole-brain imaging at cellular resolution (*Vladimirov et al., 2014*). By analyzing the relationship between spontaneous brain activity and spontaneous behavior (*Freeman et al., 2014*), we generated whole-brain activity maps of neuronal and neuropil structures that correlated well with the observed locomotor patterns. We revealed anatomically structured neural populations in the hindbrain with activity fluctuating on slow

timescales similar to the periods of directional locomotion that characterize spatiotemporal behavioral patterning. Subsequent circuit perturbations established a link between these populations and self-generated swim statistics. Finally, we showed that these cells are composed of two glutamatergic clusters and two GABAergic clusters that potentially form a mutually inhibitory circuit motif. We suggest that these neuronal populations are part of a network that confers temporal structure to spontaneous behavior to optimize innate spontaneous exploration.

## Results

### Fish exhibit a structured spatiotemporal pattern of spontaneous swimming

We mapped the statistical structure of larval zebrafish swim patterns in homogenous environments providing no explicit sensory input to ask if we could detect structural features in spontaneous behavior (*Figure 1A*, *left*). Because larval zebrafish swim in discrete swim bouts (*Figure 1A*, *right*; on average one bout every 1.22 ± 0.16 s, mean ± SEM across fish), the behavior could be partitioned into a punctuated series of swim bout locations and turn angles (*Figure 1B*). We observed that fish do not randomly choose a turning direction, but rather string together repeated turns in one direction before stochastically switching to a chain of turns in the other direction (*Figure 1B,C*; *Video 1*; overall turn angle distribution shown in *Figure 1D*). We quantified this observation by constructing a null hypothesis that the chains of ipsilateral turns arise by chance from a fish choosing randomly to turn left or right independent of turn history. In real fish, correlations in turn direction resulted in an increase in cumulative signed turn direction after a switch in turn direction. This increase was significantly different, for chains of five turns, from that of a model fish swimming left and right randomly (*Figure 1E*; *Figure 1—figure supplement 1A*; *Figure 1—source data 1*; see 'Materials and methods'). Furthermore, histograms of streak length showed that long streaks were significantly more prevalent in the real data than in the model fish, up to streaks of at least 15 turns (*Figure 1F*; *Figure 1—figure supplement 1B*). We conclude that freely swimming fish spontaneously chain together turns biased in the same direction for approximately 6 seconds on average (assuming 1.2 seconds/bout) and much longer in some periods (45% of bouts are in streaks of 5 bouts or longer; 14% in streaks of 10 bouts or longer; see *Figure 1—figure supplement 1B,C*), significantly deviating from a random walk (*Codling et al., 2008*).

### Structure of spontaneous locomotion in fictively behaving zebrafish

We sought to characterize the neural basis of these spontaneous behavioral sequences. In order to do this, we first determined if similar temporal turning structures could be observed in a paralyzed fish preparation compatible with live microscopy (*Figure 1G*). Fictive turns were decoded using electrical recordings from peripheral motor nerves and an algorithm that compares the signals on the left and right electrodes after normalizing each channel to account for signal strength differences (see 'Materials and methods') (*Ahrens et al., 2013*). We verified the accuracy of our turn decoding algorithm using two complementary strategies, detailed in 'Materials and methods' and *Figure 1— figure supplement 2*. First, we used a three-electrode setup to verify that fictive swims that were decoded to have a small angular component indeed consisted of traveling waves along the tail, and that motor events decoded to have a large angular component consisted of concurrent signals along the ipsilateral rostrocaudal extent of the tail during the initial burst (*Figure 1—figure supplement 2A–F*), as in freely swimming fish (*Mirat et al., 2013*). Second, we used weakly paralyzed fish to record fictive signals and residual tail motion simultaneously, and found an accurate match between decoded fictive turn direction and tail movement (*Figure 1—figure supplement 2G–K*). We also verified that the fictive swim events did not contain struggles or Mauthner-mediated escapes (*Figure 1—figure supplement 3*). Additionally, although not essential to the subsequent analyses, by matching the distributions of fictive turn angles to the corresponding distributions of freely swimming fish, we generated approximate two-dimensional trajectories in virtual space from the fictive recordings (*Figure 1H*), as described previously (*Ahrens et al., 2013*).

   We confirmed that fictively behaving zebrafish exhibit behavioral sequences similar to freely swimming zebrafish, with similar chains of unilateral turns (compare *Figure 1H–L* to *Figure 1B–F*, *Figure 1—figure supplement 1B–C*). While the fictive swim frequency was slightly slower relative to

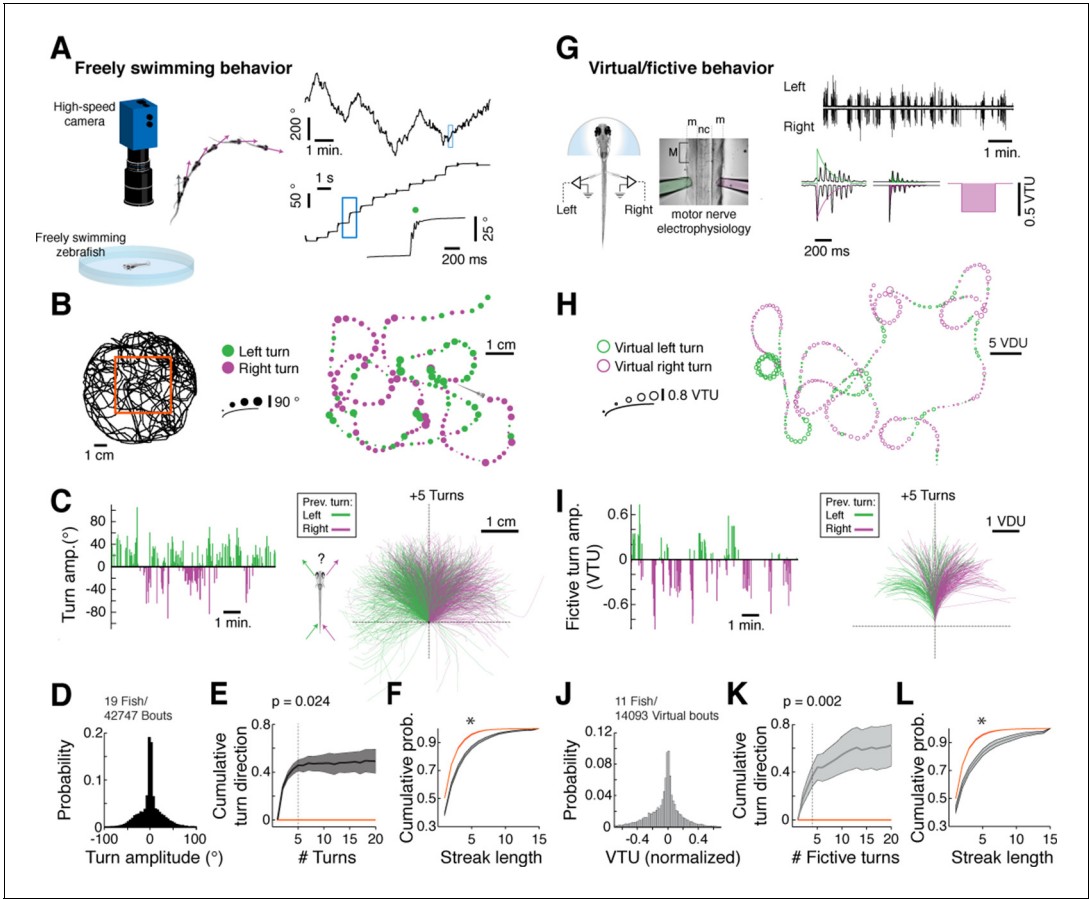

**Figure 1.** Spontaneous orienting behavior is governed by switches in turn state. (A) As fish explore a homogeneous environment, heading direction (*purple vectors*) over time is recorded with a high-speed camera. Fish execute discrete spontaneous turns (*top right*, showing heading direction over time) that comprise sequences of turns biased in the same direction (*middle right*). The size and color of the spot (*bottom right*) denote the magnitude and direction of the underlying turn, respectively. (B) Plot of a swim trajectory taken from a much longer recording (from within red box, *left*). Turn direction is encoded by color (left turns in green; right turns in magenta). Dots are positioned at the points in the trajectory where turns were executed; dot size is proportional to turn angle. Note the chains of left and right turns that confer a characteristic slaloming shape to the swim trajectory. (C) *Left*, turn states can also be visualized by plots of turn amplitude over time, colored according to turn direction. Notice that the fish tends to turn in streaks. *Right*, when swim trajectories are triggered, rotated, aligned, and color-coded according to the direction of each preceding turn, it is again evident that the previous turn biases the future trajectory of the animals in the same direction. (D) Histogram of turn amplitudes from 42,747 swim events across 19 fish. The overall turn distribution is symmetric. (E) Quantification of average turn history-dependence. After a left -> right or right -> left switch event, fish tend to turn with a bias in the same direction for 5 swim bouts (p=0.024 for the change in cumulative turn direction, signed rank test compared to a randomly turning model fish, shown in *red*). N = 19 fish. Shaded error is SEM across fish. (F) Cumulative histogram of streak length (number of turns in the same direction before a switch) for 19 fish, *black*, compared to randomly turning fish, *red*. (*) p<10⁻⁵ rank sum test. Shaded error is SEM across fish. (G) Turns can also be decoded from electrophysiological recordings from peripheral motor nerves in paralyzed fish. Turn direction and amplitude are calculated by subtracting the normalized power of recorded bursts in the left channel (green electrode) from bursts in the right channel (pink electrode), weighing the start of a burst more than the end (see exponential filters, *right*, and 'Materials and methods'). (m) muscle; (nc) notochord; M (myotome); VTU (virtual turn unit). (H) Sequences of decoded virtual turns and virtual swim distances (sum of left and right fictive channels, virtual distance units [VDU]) can be used to plot virtual swim trajectories. The pattern of unidirectional sequences observed in freely swimming fish is conserved in fictively swimming fish. (I) Fictive turn amplitude and trajectory history plot for the data in (G) and (H). (J) Histogram of fictive turn amplitudes from 14,093 swim events in 11 fish. (K) Across animals, fictively swimming fish tend to turn with a bias in the same direction for 4 swim bouts after a change in turn direction (p=0.002 for the change in cumulative turn direction, signed rank test compared to a randomly turning model fish, *red*), although many chains persist for much longer. N = 11 fish. Shaded error is SEM across fish. (L) The cumulative probability distribution of fictive streak length is also significantly different from a randomly turning model fish. (*) p<10⁻⁴, rank sum test. N = 11 fish. Shaded error is SEM across fish.

The following source data and figure supplements are available for figure 1:

**Source data 1.** Behavioral data from freely swimming larval zebrafish, with analysis code.

**Figure supplement 1.** Analysis of free and fictive turn states.

*Figure 1 continued on next page*

*Figure 1 continued*

**Figure supplement 2.** Fictive swimming is a reliable readout of intended locomotion.

**Figure supplement 3.** Fictive swims are not struggles or startles.

**Figure supplement 4.** Comparison of free and fictive swimming statistics.

freely swimming conditions ($\text{mode}_{fictive}/\text{mode}_{free}$ = 1.4; $\text{mean}_{fictive}/\text{mean}_{free}$ = 2.4; $\text{median}_{fictive}/\text{median}_{free}$ = 1.9), the large overlap between the histograms of inter-bout intervals (IBIs) (***Figure 1—figure supplement 4A–C***) suggests that fictive behavior is sufficiently similar to that of freely swimming fish to allow for analysis of concurrent neural signals.

## Recording brain activity during spontaneous fictive locomotion

We used light-sheet microscopy to record neural activity in most neurons in the brain during spontaneous fictive locomotion. Using transgenic zebrafish expressing the genetically encoded calcium indicator GCaMP6f or GCaMP6s (***Chen et al., 2013***) in most neurons ('Materials and methods'), and a dual-laser light-sheet microscope capable of scanning the entire brain without exposing the retina to the laser beam (***Vladimirov et al., 2014***) (***Figure 2A***), we captured whole-brain neuron-resolution activity at 1.87 ± 0.14 (mean ± SD over all recordings) brain volumes per second during spontaneous behavior (***Video 2***). To map the relationship between whole-brain activity and behavioral sequences, we constructed a representation of swimming events that allows for nonlinear relationships between neuronal activity and the strength and direction of turns. To generate this representation, fictive recordings were transformed into distinct behavioral events (***Ahrens et al., 2013***) (***Figure 2B***, left; 'Materials and methods'), each associated with a particular point in a two-dimensional 'behavioral tuning space', analogous to a visual receptive field. In this space, angle represents turning direction, and radial distance represents the strength of the motor event (***Figure 2B***, middle). By regressing whole-brain activity against this representation of behavior (***Figure 2B***, right; 'Materials and methods') (***Portugues et al., 2014***; ***Miri et al., 2011***), signals from individual voxels (or neurons) were thus described with a tuning field over the behavioral space (***Figure 2C***; two example neurons, one tuned to left and the other to right turns).

## Whole-brain maps reveal neural representations of spontaneous behavior

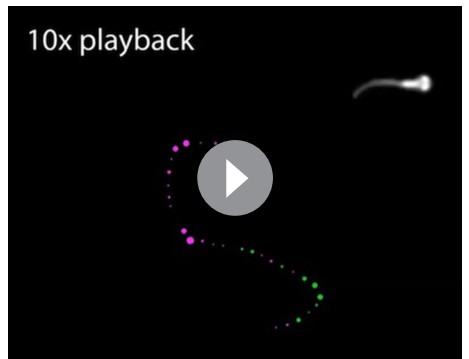

**Video 1.** Spontaneous freely swimming behavior video of a freely swimming larval zebrafish, with turn direction coded by color (green = left; magenta = right) and turn amplitude coded by the size of the dot. Fish was masked from background and smoothed to produce the representation on the top right.

To match neural activity to the pattern of spontaneous turning, we generated whole-brain activity maps for individual fish by color-coding each voxel for preferred angle (***Figure 2D***; ***Videos 3,4***; 'Materials and methods'; analysis code and example data available online). We encoded the predictability of the response, or $R^2$, with brightness so that brighter colors mean more significant correlations to behavior (***Figure 2D–F***). These maps revealed the organization of behavioral tuning in both neurons and neuropil. Neurons linked to spontaneous swimming were located in diverse areas of the brain (***Figure 2D–G***). Anatomical consistency of the functionally identified regions was evaluated using a nonlinear volume registration algorithm (***Portugues et al., 2014***) that aligned seven brains based only on anatomy; this analysis revealed that, across fish, the most spatially conserved functionally defined cell clusters were in a

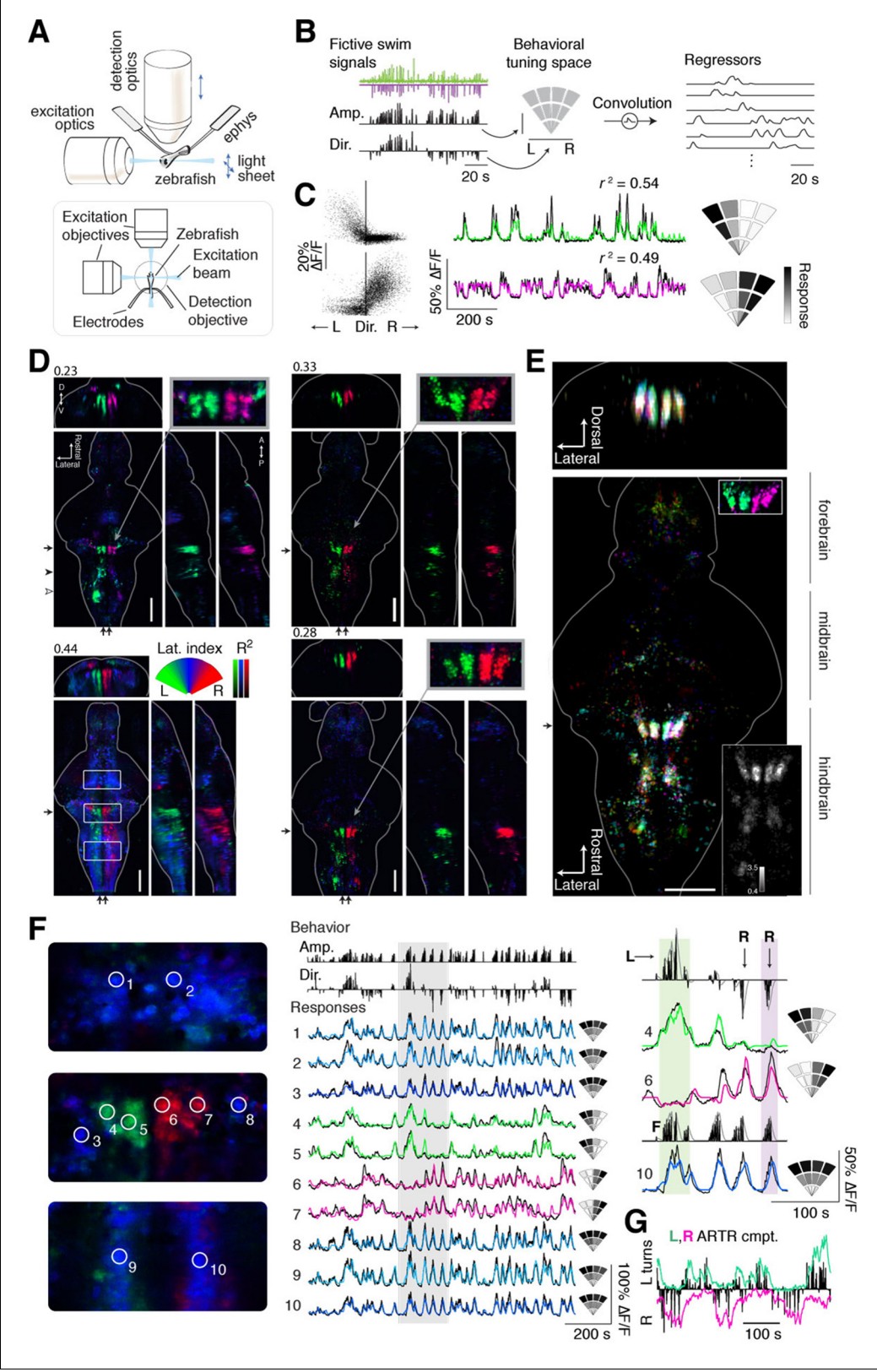

**Figure 2.** Whole-brain analysis identifies neural structures correlated with turning behavior. (**A**) Schematic of experimental paradigm for fictive swimming combined with light-sheet imaging ('Materials and methods'). (**B**) Schematic of analysis technique. Left: First, fictive swim signals are converted into measures of swim amplitude
*Figure 2 continued on next page*

*Figure 2 continued*

('Amp') and turning direction ('Dir' for laterality). Middle: Next, amplitude and laterality are mapped onto the vertical and horizontal axes of a 2D space. This space is tiled with 12 basis functions, each representing a region in this 2D behavior space, now defined in polar coordinates ('Materials and methods'). Contours are shown for clarity; actual basis functions overlap by 50%. Right: The signal from each bin is convolved with an impulse response function to generate a regressor; an example subset of regressors is shown. (C) Brain activity is regressed against the regressors constructed in (B) to generate a behavioral tuning function for every voxel. Voxels of two example neurons are shown here. Left, relationship between turn laterality and neural response for the two example neurons, each dot is a time point. Middle, time series from the same two example neurons. Black line, ΔF/F; colored line, prediction of best-fitting model (see panel B). Right, behavioral tuning for the same two neurons, given by regression coefficients, using the analysis described in panel B; grayscale ranges from $10^{th}$ to $90^{th}$ percentile of the coefficient weights. (D) Behavioral tuning maps across the brain derived from fitting every voxel with the regressors described in panel B, for four representative fish. Calcium indicators are either localized in cytoplasm (left two fish) or in the nucleus (right two fish). The dorsal view is a maximum intensity projection over the whole brain; the side and front views are taken from a maximum intensity projection of 21 slices (~10 μm) along the medial-lateral axis and rostral-caudal axis, respectively. Numbers above each panel indicate the $R^2$ value at which the color map saturates (maximum $R^2$ value is higher), color maps start at $R^2 = 0$. Arrows in each panel represent the centroid position of these slices for the frontal view (*top*) or side view (*right*). Solid arrowhead: diffusive correlated region in rhombomeres 4–6. Open arrowhead, inferior olive. Scale bar, 100 μm. D, dorsal; V, ventral; A, anterior; P, posterior. (E) Registered map from seven different fish (nuclear localized GCaMP6f) to a standard brain. Each fish is encoded by a different color; brightness represents $R^2$. Bottom, top-down maximum intensity projection (along the dorsal-ventral axis). Top, front projection as in d, with the centroid of the slice indicated by the arrow in bottom panel. Top *right* inset, ARTR region across fish in the standard brain, but with color representing laterality as in panel d, showing consistent tuning across animals. *Bottom right inset*, a measure of stereotypy in location of functionally identified neurons across the 7 fish. Intensity represents the standard deviation divided by the mean of $R^2$ (thresholded at 0.04). Scale bar, 100 μm. (F) Example ΔF/F traces from regions of interest (ROIs) in panel (D) (left bottom, white boxes). Left, top to bottom: midbrain, ARTR, and caudal hindbrain. Middle, top, signals of swim amplitude (Amp.) and turn laterality (Dir.). Black bars represent several individual swim events. *Bottom*, ΔF/F from ROIs in the left panels. Right, enlarged view of gray region in middle panel. L,R,F stand for left turns, right turns and swim amplitude, respectively. Responses from ROIs 1–3 and 8–10 show tuning to swim amplitude; ROIs 4,5 to left turns, and ROIs 6,7 to right turns. (G) In addition to single cells, activity of left and right populations derived with ICA (*Figure 2—figure supplement 1C–E*; bottom-right fish of *Figure 2D*) tracks turning behavior.

The following figure supplements are available for figure 2:

**Figure supplement 1.** Alignment of functional brain maps in fish expressing calcium indicators in the cytosol.

**Figure supplement 2.** Recovering the ARTR using supervised and unsupervised methods.

**Figure supplement 3.** Dynamics of ARTR activity during behavior.

---

region in the anterior hindbrain (*Figure 2E*; *Figure 2—figure supplement 2*). This region is composed of two bilaterally symmetric clusters of cells on either side of the midline. In a previous study (*Ahrens et al., 2013*), this area was identified by brain activity alone – without behavioral readout – and termed the hindbrain oscillator (HBO). The dynamics of these cells were tightly coupled to the direction of turning and highly antiphasic, such that the majority of the time, cells in only one hemisphere were active (*Figure 2F,G*). The location and dynamics of these cells, which we call the anterior rhombencephalic turning region (ARTR), named according to its salient anatomical and functional properties, was additionally verified using two-photon imaging during fictive behavior (*Figure 2—figure supplement 1*). We manually counted the numbers of neurons in the functional brain maps and found 60 ± 7 neurons in each medial cluster and 33 ± 2 neurons in each lateral cluster (mean ± SEM, 8 fish). To visualize the relationship between ARTR dynamics at the time of turns, we aligned the neuronal activity traces to individual turns that were followed by five seconds of no turns. At the time of a turn, the ipsilateral ARTR was activated, and after a rise in calcium signal over about 2 s, decayed to baseline slowly on a timescale of 5–10 s (*Figure 2—figure supplement 3*).

To analyze the wider anatomical features of these maps across fish, we used Z-Brain (*Figure 3— figure supplement 1*) (*Randlett et al., 2015*) to register together and average multiple brain

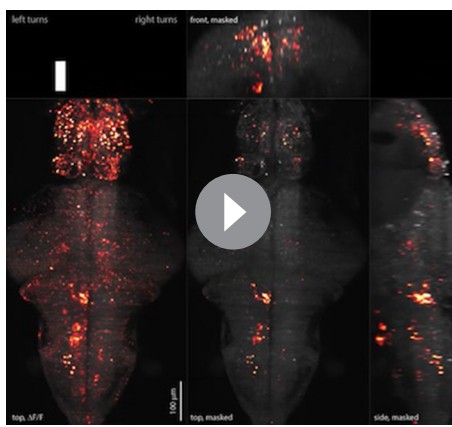

**Video 2.** Whole-brain imaging during spontaneous fictive behavior recorded in the light-sheet virtual reality setup. *Left*: top projection of whole-brain ΔF/F. *Top left*: behavior represented by left and right turn amplitude. *Right:* projections of whole-brain ΔF/F, where the brain has been masked by the $R^2$ volume, so that each voxel represents ΔF/F x $R^2$, emphasizing neural activity in the regions identified by the regression analysis.

volumes. These averaged functional brain maps (*Figure 3A–B*) revealed that the most prominent directionally tuned neurons were located in the hindbrain, where most tuning was ipsilateral to turning direction. These prominent directionally tuned neurons were found in the ARTR in rhombomeres 2–3, diffusely distributed in rhombomeres 4–6 (Rh4-6), in the inferior olive (IO), and in the vicinity of and overlapping with the reticulospinal system (*Figure 3C,E*). Weaker and less directionally tuned signals were observed in the torus longitudinalis, the habenula, the preoptic area and pretectum, the cerebellum (Cb, *Figure 3D*), and the midbrain tegmentum, including the area containing the nucleus of the medial longitudinal fasciculus (nMLF, *Figure 3E*, *Figure 3—figure supplement 2*).

These maps outline areas across the brain that are active during spontaneous locomotion, but which of these areas may underlie the slow structure in directional swimming? The strongest directional tuning was found in the ARTR, Rh4-6, and IO. Directly comparing the directional tuning of the ARTR and Rh4-6, however, revealed that the ARTR was more directionally tuned (*Figure 3—figure supplement 3A*). In addition, the ARTR was more predictive of future turn direction (*Figure 3—figure supplement 3B*). These observations suggest that the ARTR may be more involved in turn patterning. In addition, we compared the temporal dynamics of the ARTR and Rh4-6 and found that the ARTR had slower dynamics than Rh4-6 (*Figure 3—figure supplement 3C*), suggesting that the ARTR may be more involved in the slow dynamics of the behavior and Rh4-6 more in direct behavioral output. Further, while the IO was strongly directionally tuned, it projects only to the contralateral Cb (*De Zeeuw et al., 1998*; *Bae et al., 2009*). While we did see directional tuning with flipped laterality in the Cb (*Figure 3D*), this signal was only weakly tuned to behavior, and therefore, the IO-Cb circuit may be less directly related to turn patterns. These results led to the hypothesis that the ARTR might generate the slowly fluctuating bias in swimming direction.

## The ARTR biases spontaneous turning

The relationship between neuronal activity in the ARTR and fictive behavior suggests that the ARTR may underlie directionality in spontaneous swimming, such that activity in the right ARTR or the left ARTR biases turning to the right or the left, respectively. To test this hypothesis, we performed unilateral lesions of the ARTR. To do this, we first functionally identified the ARTR at the single-cell level in each fish, using two-

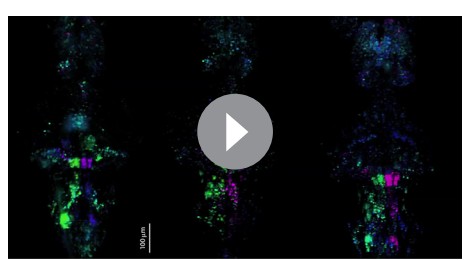

**Video 3.** Analysis of imaging data Computational brain maps of the voxel-wise tuning to the laterality of turns. Green signifies tuning to left turns; magenta to right turns; brightness codes for $R^2$ of the model fit. Same data as *Figure 2* but represented in three dimensions.

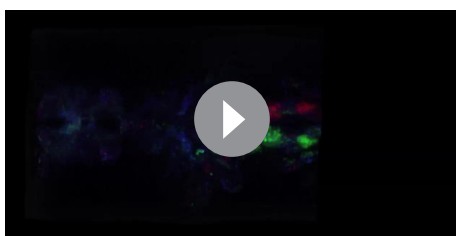

**Video 4.** 3D representation of Z-Brain atlas.

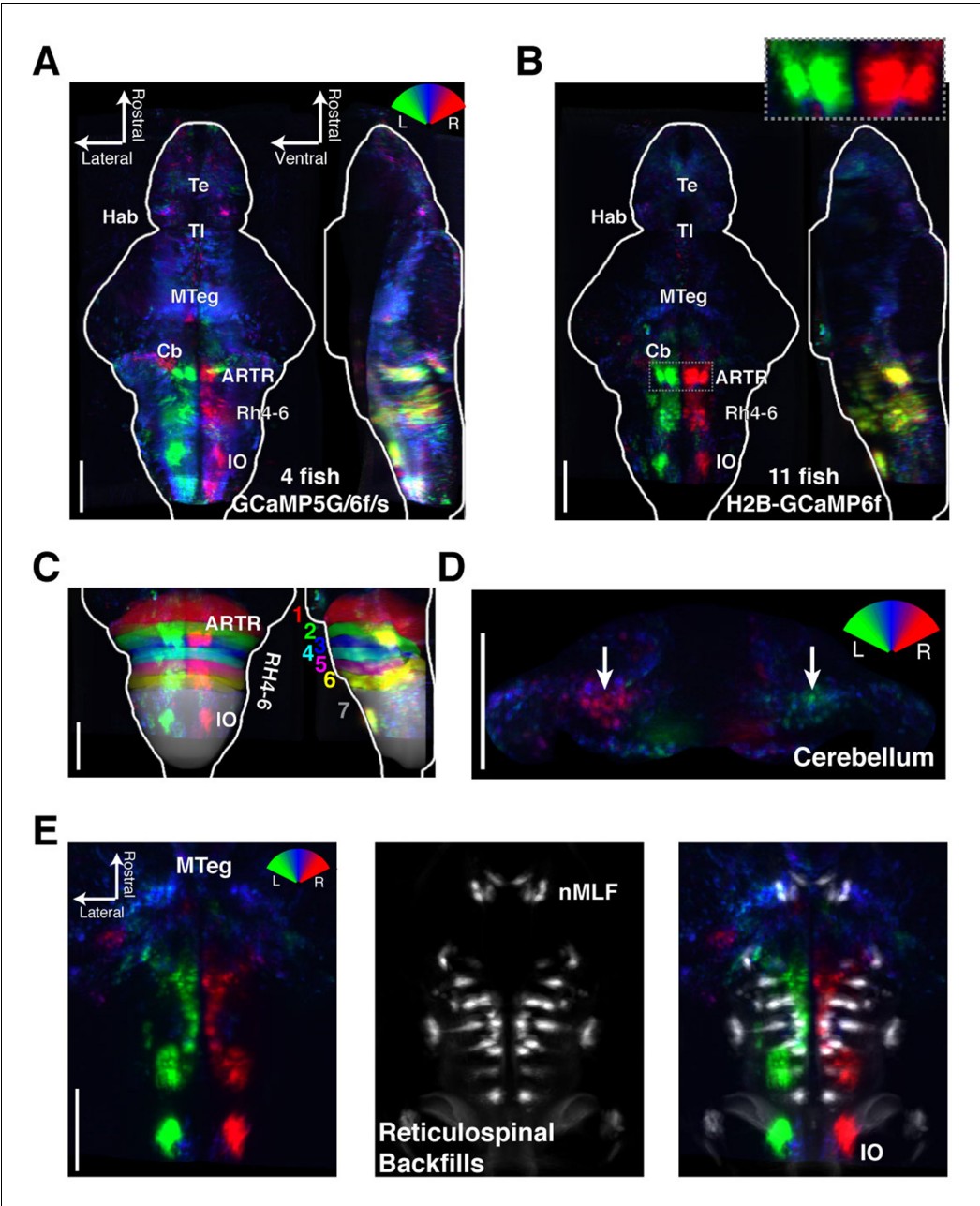

**Figure 3.** Functional anatomy of brain regions correlated with spontaneous behavior. Activity patterns consistently observed across fish highlighted by registering multiple fish to the Z-Brain atlas and averaging functional signals (see Supplementary methods). (A) Average functional stack resulting from *Tg(elavl3:GCaMP6f)* (N=2), *Tg(elavl3:GCaMP5G)* (N=1) and *Tg(elavl3:GCaMP6s)* larvae (N=1). Color represents tuning to fictive turning as in *Figure 2*. (B) Average functional stack resulting from *Tg(elavl3:H2B-GCaMP6f)* larvae (n=11). (C–E) Anatomical analyses of the average *Tg(elavl3:H2B-GCaMP6f)* maps in (B). (C) The positioning of hindbrain within the rhombomeres. (D) Untuned and some more weakly direction selective signals observed in the cerebellum. (E) Virtual colocalization comparing the position of ventral hindbrain and midbrain tegmentum (M-Teg) signals with the reticulospinal system. $R^2$ = 0...0.12 (A–C), 0...0.06 (D–E). Scale bars, 100 μm. (Te) telencephalon; (Hab) habenula; (Tl) torus longitudinalis; (Cb) cerebellum; (ARTR) anterior rhombencephalic turning region; (Rh4-6) rhombomeres 4-6; (IO) inferior olive; (nMLF) nucleus of the medial longitudinal fasciculus.

The following figure supplements are available for figure 3:

**Figure supplement 1.** Registering brains to the Z-Brain atlas.

**Figure supplement 2.** The nMLF is correlated with swim amplitude but not direction.

*Figure 3 continued on next page*

*Figure 3 continued*

**Figure supplement 3.** Comparison of ARTR and Rh4-6 dynamics, tuning, and predictiveness of future behavior.

photon imaging and fast analyses of correlated activity patterns ('Materials and methods'). This enabled us to use targeted two-photon laser ablation to lesion one side of the ARTR while keeping the other side intact (19 ± 6 cells [mean ± SD] in the medial cluster, i.e. about 32% of cells of one medial cluster; 'Materials and methods'). Freely swimming behavior was quantified before and after unilateral lesions of the ARTR. We found that post-ablation, fish turned relatively more often toward the direction of the intact half of the ARTR (shift in turn direction to intact side: 18 ± 4%, mean ± SEM; *Figure 4A–B*) (left ARTR ablation: N = 6 fish, p=0.031; right ARTR ablation: N = 7 fish, p=0.031; sham ablation: N = 5, p=0.438, paired signed rank test), suggesting that the ARTR is involved in generating directionality bias in spontaneous turning behavior.

Do these ARTR lesions affect the turn bias, or simply the ability of the animals to turn? We examined the magnitude of turns – independent of relative frequency – pre- and post-ablation. In contrast to the effect of reticulospinal neuron ablation (*Orger et al., 2008*; *Huang et al., 2013*; *Kimmel et al., 1982*), after ARTR ablation, the magnitude of turns to either the ablated or intact side remained unchanged (*Figure 4C*, p=0.147, ablated side; p=0.127, intact side, N = 13 fish, paired signed rank test). These results suggest that the ARTR is involved in regulating the choice of turning direction, rather than mediating the actual turn kinematics.

We also tested whether the ARTR contributes to temporal correlations in turn direction. For each fish, before and after ablation, we constructed a model fish consisting of a history-independent process in which every turn was randomly chosen to be to the left or to the right. Since the overall turn bias contributes to the distribution of streak lengths, the bias of each model fish must be matched to that of each real fish. Thus, the direction of every turn in each model fish effectively results from a biased coin flip, with the bias matched to each corresponding real fish (e.g. 45% right turns and 55% left turns). Under the hypothesis that the ARTR contributes to temporal correlations in turn direction, the streak length distribution of a post-ablation real fish should be closer to that of its matched model fish than a pre-ablation real fish to its matched model fish. We found that this was indeed the case (*Figure 4D–F*). Specifically, comparing the goodness-of-fit between the matched biased random walks and the observed turn sequences ('Materials and methods'), we found a significant reduction in the normalized root-mean-square error over fish after ablation (*Figure 4G*, p=0.027, paired signed rank test, N = 13 fish). These results indicate that turn sequences in lesioned fish are more similar to strings of biased coin flips than are turn sequences in intact fish, providing evidence that the ARTR is part of a circuit that implements a temporally correlated process.

To further test whether the ARTR biases turn direction, we optogenetically stimulated the ARTR in fish expressing both GCaMP6f and the excitatory opsin ReaChR (*Lin et al., 2013*) under the *elavl3* promoter ('Materials and methods'). We first imaged the area of the ARTR using 930 nm light (near the peak two-photon excitation wavelength of GCaMP6f) and functionally identified ARTR neurons. Next, we selected an ROI over 15 to 20 ARTR neurons and stimulated these neurons using 1050 nm light (near the peak two-photon excitation wavelength of ReaChR). Stimulating ARTR cells caused an increase in the number of turns in the direction ipsilateral to the stimulated ARTR region (*Figure 4H–K*; 29 ± 9%, mean ± SEM, shift in turns to the stimulated direction; left stimulation, p=0.004; right stimulation, p=0.002, paired signed rank test, N = 7 fish), independent of whether the medial clusters or lateral clusters were chosen for stimulation (*Figure 4—figure supplement 1A*). These results are corroborated by electrical stimulation experiments (*Figure 4—figure supplement 1B–D*), demonstrating that the ARTR is functionally connected to downstream motor circuitry and that activity in the ARTR influences turn direction. In contrast to our ablation results, the magnitude of turns during stimulation of the medial clusters tended to increase (*Figure 4L*, *Figure 4—figure supplement 1A*, left stimulation, p=0.010; right stimulation, p=0.036, paired signed rank test, N = 7 fish), indicating that either the ARTR is able to bias turn magnitude in addition to turn direction, or that there is a spillover of artificial ARTR stimulation to downstream circuits.

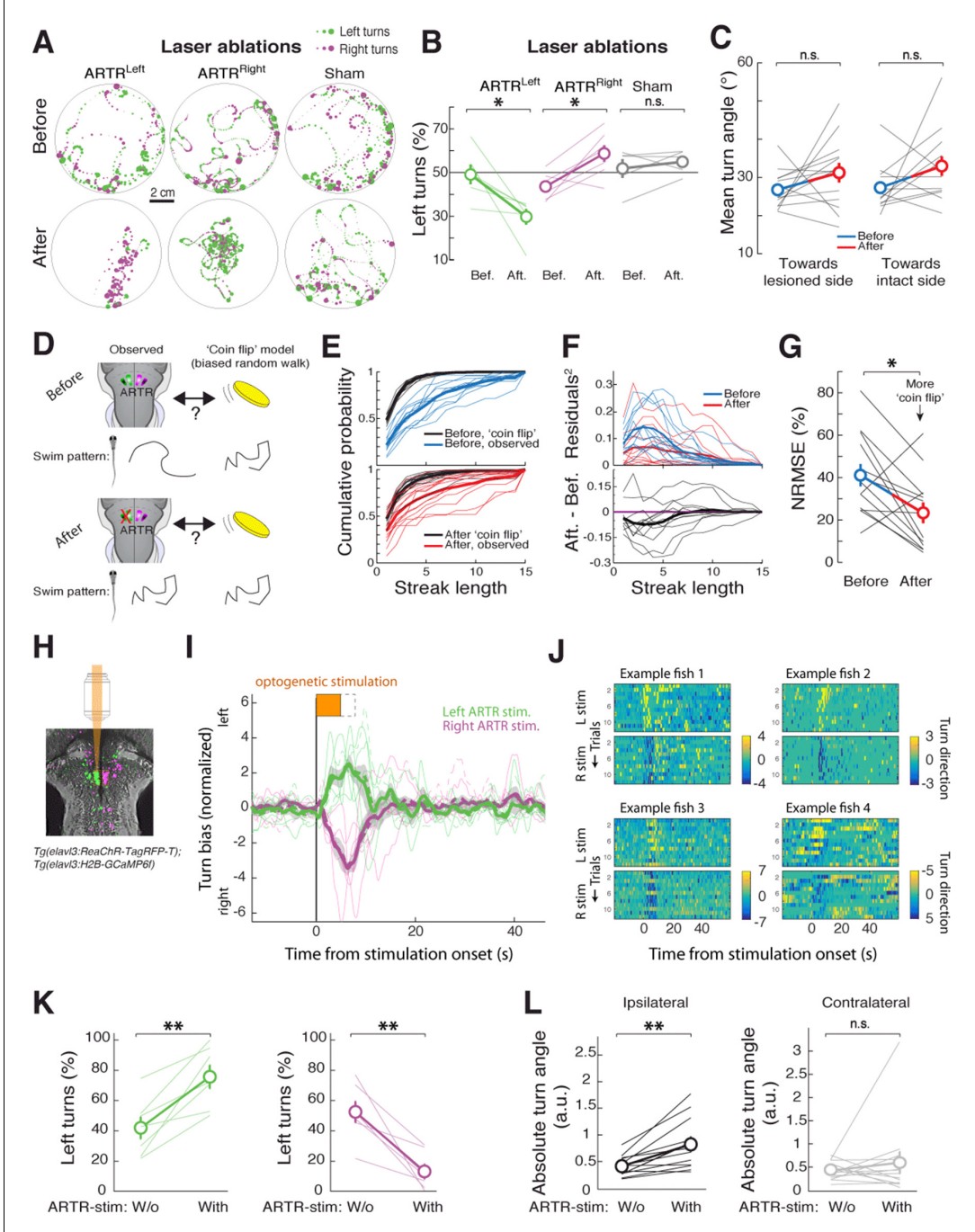

**Figure 4.** The ARTR biases turn direction. (**A,B**) Unilateral laser ablation of a subset of cells in the ARTR reduces ipsilateral turns. (**A**) Example swim trajectories, shown over a subset of the duration of the experiment, and (**B**) summary of turning behavior before and after laser ablation of cells in the left medial cluster (*green*) or right medial cluster (*magenta*) of the ARTR. *Gray*, data from sham ablations of hindbrain neurons outside of the ARTR. Only events occurring more than 1 cm away from the wall were analyzed. (**C**) Mean turn angle to the lesioned or intact side, before and after ablation. Although the relative frequency of turns to the ablated side decreases (**B**), fish remain capable of executing turns of normal magnitude to the lesioned side, with no significant difference in mean turn angle between pre- and post-ablation conditions. (**D**) Schematic of hypothesized changes to swim structure, assuming that the ARTR is involved in setting correlational patterns. Before and after ablation, turn patterns will be compared to a 'coin flip' model that emits turns to the left and right randomly but with some bias equal to the observed data. (**E**) Empirical cumulative distribution functions (CDFs) of streak length before (*blue, top*) and after (*red, bottom*) ablation, compared to model fish executing turns at random without history dependence but with overall turn bias matched to each individual fish (*black*). Streak length post-ablation appears distributed more like 'coin flips'. (**F**) *Top*, quantification of the squared residuals between each individual fish CDF and its matched 'coin flip' CDF before (*blue*) and after (*red*) ablation. *Bottom*, the difference between each respective before and after curve reveals a shift toward the 'coin flip' distribution for the majority of fish. (**G**)

*Figure 4 continued*

Summary of the normalized root-mean-square error (NRMSE) quantifying goodness-of-fit between the observed streak distributions and their matched random model distributions. After ablation, turning becomes more 'coin flip'-like and thus history dependence is reduced. (H–L) Optogenetic stimulation of the ARTR elicits ipsilateral turn biases. (H) The ARTR was functionally identified in double transgenic fish *Tg(elavl3:H2B-GCaMP6f;elavl3: ReaChR-TagRFP-T)* and a medial ARTR cluster was unilaterally stimulated. *Gray*, expression of ReaChR-TagRFP-T; *green and magenta*, functionally identified ARTR from this example fish based on correlational map. (I) Ipsilateral turn bias increases during optogenetic stimulation (solid lines, 5 s stimulation; dotted lines, 8 s stimulation, N = 7 fish, 'Materials and methods'). (J) Results from example fish show the reproducibility of stimulation effect across trials. Turn direction is normalized to time-averaged turn direction pre-stimulation. (K) Summary of the change in bias quantified for each fish, showing that optogenetic stimulation results in a bias toward ipsilateral turns. (L) Summary of the change in absolute fictive turn angle during stimulation, showing that ipsilateral turn angle increases and contralateral turn angle remains unchanged. n.s., no significance; (*) p<0.05; (**) p<0.01 (paired signed rank test). All error bars are mean ± SEM across fish.

The following figure supplement is available for figure 4:

**Figure supplement 1.** Detailed analysis of ARTR stimulations.

## ARTR neurotransmitter identity and morphology

Given these functional observations of ARTR activity and its effect on behavior, how might ARTR connectivity give rise to its activity patterns and cause it to influence motor output? To address this question, we investigated the projection patterns of ARTR cells to look for evidence of putative intrinsic connectivity and putative connectivity to premotor neurons.

Registration to the Z-Brain atlas suggested a unique distribution of neurotransmitter identities within the ARTR, which comprises a pair of medial and lateral clusters in both hemispheres. In the Z-Brain atlas, the medial and lateral clusters mostly overlapped with glutamate and GABA markers, respectively (*Figure 5—figure supplement 1A,B*). To verify this overlap, we functionally identified the ARTR in double transgenic lines with GCaMP6f and red fluorescent labels for either glutamatergic (*vglut2a*) or GABAergic (*gad1b*) neurons and matched it with its neurotransmitter phenotype (*Figure 5A*, *Figure 5—figure supplement 1C,D*, 'Materials and methods'). In this way, the medial clusters of the ARTR were identified as being glutamatergic, and the lateral clusters as being primarily GABAergic (*Figure 5A middle* and *right*, respectively).

How might such an arrangement of excitatory and inhibitory neurons lead to the activity patterns observed in the ARTR? The strongly antiphasic activity patterns and the presence of excitatory and inhibitory clusters suggests underlying mutual inhibition, i.e. when one side of the ARTR is active, the other side is suppressed. To probe for such an architecture, we used a combination of calcium imaging and photoactivateable GFP (PA-GFP) (*Patterson, 2002*; *Ruta et al., 2010*; *Datta et al., 2008*). After functionally identifying the ARTR using a nuclear-localized calcium indicator (H2B-GCaMP6f), we photoactivated PA-GFP in a subset of ARTR cells and traced their projections. Indeed, tracing PA-GFP-activated neurites from the lateral GABAergic clusters revealed projections reaching across the midline toward both contralateral ARTR clusters (*Figure 5B* and *Figure 5—figure supplement 1E–G*). In contrast, we did not find evidence for neurites from the medial glutamatergic clusters crossing the midline (*Figure 5B*, *bottom right*). Although these tracing studies cannot prove whether there exists synaptic connectivity between the ARTR clusters, they are suggestive of a mutually inhibitory circuit motif, which could mediate the antiphasic activation necessary for patterning directional motor output.

To investigate potential connectivity between the ARTR and downstream premotor circuitry, we again combined functional imaging with anatomical tracing, this time using PA-GFP with a red fluorescent calcium indicator (jRCaMP1a [*Dana et al., 2016*]), which improved our ability to trace more distal neurites. Photoactivation of the medial ARTR cluster revealed that while some labeled neurites terminated in the IO (*Figure 5C*), projections also descended into the MLF and were visible adjacent to reticulospinal neurons RoV3 and RoM3 (*Figure 5D*), of which RoV3 has been shown to be directionally tuned during the optomotor response (*Orger et al., 2008*). Whether this projection represents a connection that would allow the ARTR to exert a turning bias through excitation of premotor neurons remains to be tested. We also looked at differences in temporal dynamics between ARTR activity and vSPN activity. We found that both sets of neurons were directionally tuned (*Figure 5—figure supplement 2A*), but that the ARTR exhibited slower dynamics in the fluorescence signal than the vSPNs

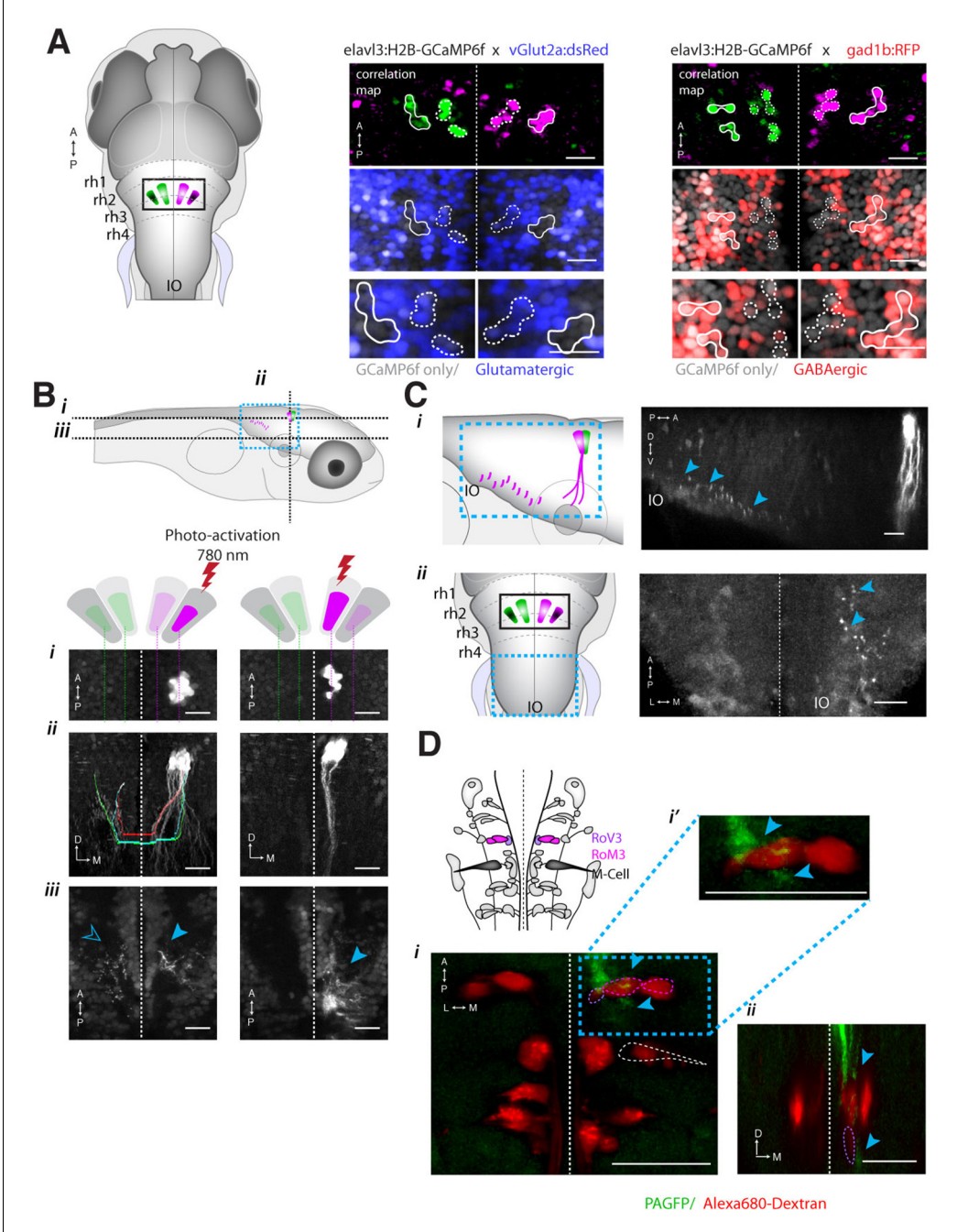

**Figure 5.** ARTR anatomy suggests mutual inhibition and connections to premotor neurons. (**A**) The medial ARTR is glutamatergic and the lateral ARTR is GABA-ergic. *Left*, Anatomical diagram showing the approximate location of the ARTR (*black box*) in rhombomeres 2–3. Green and magenta represent clusters influencing left and right turns, respectively. *Right*, The ARTR was functionally identified in two-photon imaging sessions ('Materials and methods') in transgenic fish expressing H2B-GCaMP6f in most neurons and a red indicator either in the glutamatergic (*left*) or in the GABAergic (*right*) neurons. Overlaying the functional maps (*top*) in which the ARTR cells are identified by correlation reveals that the center clusters (dotted outlines) are glutamatergic and the lateral clusters (solid outlines) are GABAergic. For the vGlut experiments, N = 25 fish; GAD experiments, N = 11 fish; one representative fish shown for each. Scale bars, 20 μm. (IO) inferior olive; (rh1-4) rhombomeres 1-4. (**B**) The lateral ARTR projects contralaterally. *Top,* Anatomical diagram showing the approximate location of the planes shown below in (i), (ii), and (iii). The ARTR was identified as in (**A**) in *Tg(H2B:GCaMP6f; α-tubulin-PAGFP)* fish and PA-GFP was activated specifically in ARTR neurons of either the lateral cluster (*left panels*) or medial cluster (*right panels*).
*Figure 5 continued on next page*

*Figure 5 continued*
Projections were traced, revealing that the GABAergic cells of the lateral cluster cross the midline (dotted white line) toward the contralateral clusters (*ii, left*). The medial glutamatergic clusters project ventrally and ipsilaterally but were not found to cross the midline (*ii and iii, right*). Solid blue arrowheads, neurites in the hemisphere ipsilateral to the activated ARTR. Open blue arrowhead, neurites in the hemisphere contralateral to the activated ARTR. (C) Cells of the medial ARTR project to the ipsilateral IO. (*i*) The ARTR was functionally identified and photoactivated as in (B) in fish expressing PA-GFP and a red calcium indicator (*Tg(elavl3:jRCaMP1a)*). *Left*, schematic of the ARTR and PA-GFP-positive neurites projecting from the medial ARTR cluster and terminating in the IO (pink). Blue dashed rectangle represents the location of the region shown on the *right*. PA-GFP positive terminals are observed in the ipsilateral IO (blue arrowheads) (*ii*) *Left*, schematic showing the location (blue dashed rectangle) of (*right*) the top-down confocal image of the terminals shown in (*i*). (D) Cells of the medial ARTR send projections to a region nearby reticulospinal neurons. *Top left,* schematic of the reticulospinal system, adapted from Orger et al. (*Orger et al., 2008*), with RoV3, RoM3 and the Mauthner cell highlighted (Mauthner cell out of plane). (i) PA-GFP positive neurites (blue arrowheads) shown nearby RoV3 / RoM3 in fish where the reticulospinal neurons were retrogradely labeled with Alexa-680-dextran (red); panel (i') shows a magnification of the boxed region. (ii) A coronal view of the two cells (RoV3/RoM3) shown in (i) and inset. Scale bars, 20 μm. A, anterior; P, posterior; L, lateral; M, medial; D, dorsal; V, ventral.

The following figure supplements are available for figure 5:

**Figure supplement 1.** Cells of the lateral ARTR are inhibitory and project to the contralateral hindbrain.
**Figure supplement 2.** Timescales of reticulospinal, ARTR and turn state correlations.
**Figure supplement 3.** The ARTR is recruited by whole-field motion.

---

(*Figure 5—figure supplement 2B–E*). Future voltage imaging or electrophysiological recordings will be needed to confirm that this difference in time constant is also true on the level of spiking.

We also tested whether ARTR neurons are exclusively activated during spontaneous turning or also during visually driven behaviors and found that the ARTR is also activated during the optomotor response (*Figure 5—figure supplement 3*), suggesting that this brain area subserves multiple behaviors.

## The spatiotemporal pattern of spontaneous swimming may improve exploration efficiency

We speculated that there might be an ethological role for the slaloming trajectories we consistently observed across fish. Because correlated directional locomotion results in a unique spatial pattern over time, we posited that it may reflect a baseline foraging strategy when external guiding cues are scarce. This idea is supported by engineering literature wherein chaotic oscillators driving autonomous agents (*Tlelo-Cuautle and Ramos-López, 2014*; *Mobus and Fisher, 1999*) produce winding trajectories that efficiently and evenly cover a space, without diffusing into faraway regions. Although the ARTR is probably not a chaotic oscillator, it does exhibit stochastic transitions between states. To study the properties of correlated trajectories produced by the ARTR, we constructed a simple phenomenological model to generate model fish trajectories, realizing that Markov models have often been used to study exploration strategies and transitions between behavioral states (*Berman et al., 2014*; *Codling et al., 2008*; *Korobkova et al., 2006*; *Gallagher et al., 2013*; *Miller et al., 2005*). This two-state Markov model (*Rabiner, 1989*), which stochastically switches between a 'left turn' state and a 'right turn' state but exhibits high probabilities of remaining in the same turn state (*Figure 6A*; *Figure 6—figure supplement 1A*, 'Materials and methods'), produced behavior similar to that of real fish (*Figure 6—figure supplement 1B,C*) based only on low-level parameters. Using this Markov model to simulate trajectories through virtual space (*Figure 6B*), we show that such a scheme covers a restricted area more efficiently than a model fish turning left and right randomly without turn history dependence, and reduces diffusion into faraway regions (*Figure 6C–F*, *Figure 6—figure supplement 1E–H*). This strategy presents two distinct advantages. First, rapid spatial diffusion may lead the fish into unknown and potentially unsafe territories; this should be prevented. Second, given this preference to remain local, covering an area efficiently in

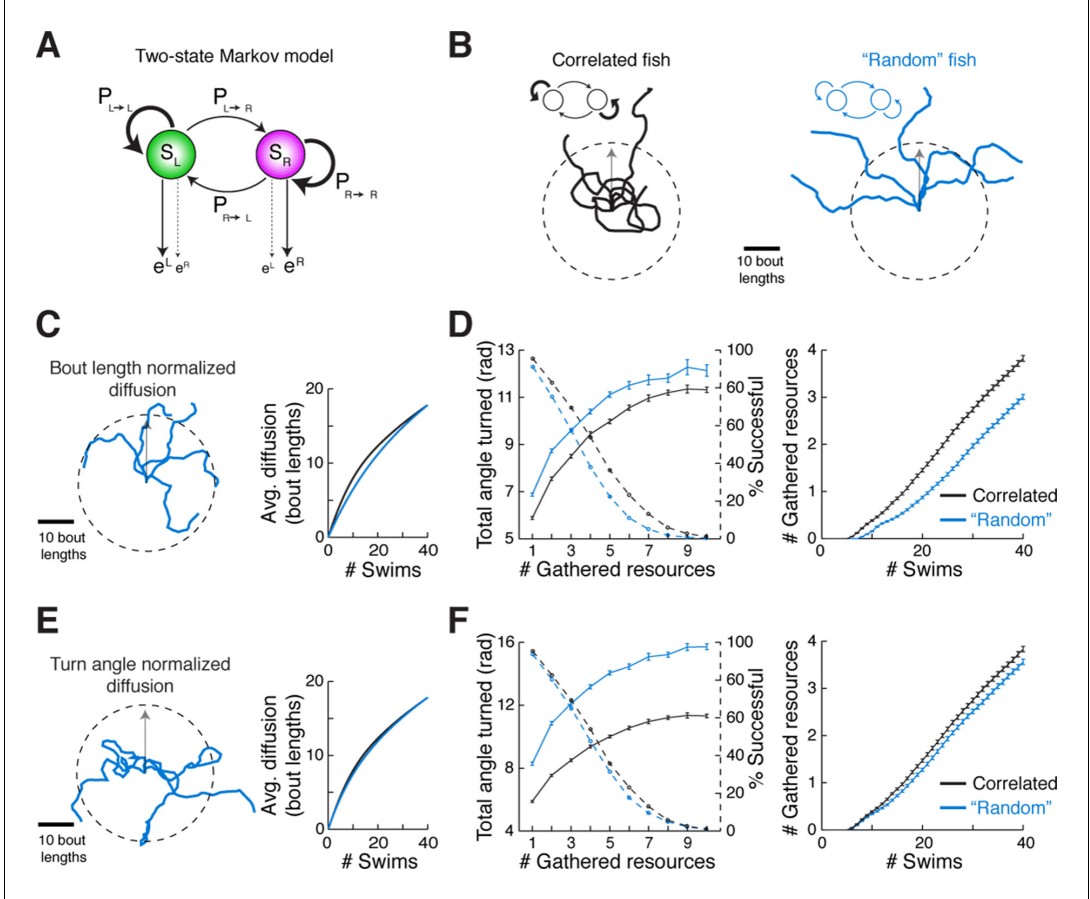

**Figure 6.** Correlated turn states may underlie efficient local exploration (**A**) Spontaneous turn states are well-characterized by a two-state Markov model (**Figure 6—figure supplement 1**). In an average model fit, fish in the left state, $S_L$, are much more likely (~90%) to turn left ($e^L$) than right ($e^R$), and vice-versa. And fish in $S_L$ or $S_R$ tend to return to $S_L$ or $S_R$, respectively, after a turn. (**B**) Left, black, Five swim trajectories generated with a Markov model matching the statistics of acquired swim data (see fish 16, **Figure 6—figure supplement 1**, $P_{transition}$ = [$P_{L \blacktriangleright L}$ $P_{L \blacktriangleright R}$ $P_{R \blacktriangleright L}$ $P_{R \blacktriangleright R}$] = [0.86 0.14 0.15 0.85]). Right, blue, five swim trajectories generated with a Markov model randomly emitting left and right turns (all $P_{transition}$ = 0.5). Notice that the unadjusted 'random' fish diffuses farther from the given starting position. The dotted circle represents the mean diffusion distance for the correlated model fish. All trajectories begin at the center of the circle and facing in the direction of the arrow. (**C**) Left, five example trajectories from the 'random' model fish after average diffusion has (right) been matched to the correlated fish by decreasing bout distance. (**D**) Plots of exploration efficiency for the 'random' model fish normalized by bout distance. Left, in this local regime, the 'random' model fish must turn more (16.9% more for 1 resource, $p < 10^{-9}$; $p = 0.004$ for 10 resources, two-tailed t-test) and (right) execute more swim bouts (21.3% fewer resources after 40 swims, $p = <10^{-9}$, two-tailed t-test) than the correlated model to collect randomly distributed virtual resources. Left, dashed lines, plots showing the proportion of simulated trajectories able to gather the indicated number of resources after 40 swims. Error bars are SEM, see **Figure 6—figure supplement 1E,F**; 'Materials and methods'. (**E**) Left, five example trajectories from the 'random' model fish after average diffusion has (right) been matched to the correlated model fish by broadening the underlying turn angle distribution. (**F**) Plots of exploration efficiency for the 'random' model fish normalized by turn angle. Left, this 'random' fish must turn much more (40.8% more for 1 resource, $p < 10^{-9}$; $p < 10^{-9}$ for 10 resources, two-tailed t-test) and (right) execute more swim bouts (7.0% fewer resources after 40 swims, $p = 9.9 \times 10^{-4}$, two-tailed t-test) than the correlated model to collect randomly distributed virtual resources. Left, dashed lines, plots showing the proportion of simulated trajectories able to gather the indicated number of resources after 40 swims. Error bars are SEM, see **Figure 6—figure supplement 1G,H**; 'Materials and methods'.

The following figure supplement is available for figure 6:

**Figure supplement 1.** Validation of two-state Markov model.

the search for food cues represents an energetically favorable program that ensures no nearby resources have been missed. Thus, we speculate that the ARTR is part of a circuit that implements a foraging strategy that discourages travel into uncertain territory, instead favoring efficient and even exploration of the local environment.

## Discussion

We uncovered an anatomically and functionally defined population of neurons that we propose is part of a circuit generating the spontaneous, patterned statistics of a directional locomotion behavior. We speculate that the function of the ARTR is to coordinate multiple successive swim bouts in order to shape trajectories on spatial scales larger than individual locomotor events. Analogous behavioral strategies must exist in other animals that explore environments much larger than themselves (*Flavell et al., 2013*; *Stephens, 1986*; *Charnov, 1976*). Thus, we expect that neural systems coordinating the transformation of multiple local actions into global actions also exist in other organisms.

The challenge in identifying neural circuits underlying such behavior lies both in the characterization of the behavior (*Stephens et al., 2008*) and in locating neural structures implementing observed behavioral schema. Spontaneously active single neurons in primates (*Okano and Tanji, 1987*) and invertebrates (*Kagaya and Takahata, 2011*) have been studied in concert with behavior, as well as neurons in invertebrates that trigger behaviors such as escape responses, exploratory limb movements and particular walking patterns (*Berg et al., 2015*; *Bidaye et al., 2014*; *Fotowat and Gabbiani, 2011*). Here, we harnessed the power of fast whole-brain imaging to describe, in detail, a nucleus in the zebrafish hindbrain influencing a simple but potentially vital behavioral algorithm that may optimize foraging when available information about the environment is scarce.

We hypothesize that the ARTR contributes to the control of correlations in turn direction according to the following mechanism. Activity on one side of the ARTR biases turn direction by subthreshold excitation of reticulospinal neurons by the medial cluster of cells. When the fish turns in the direction where the ARTR is active, a motor copy feeds back on the ipsilateral ARTR and re-activates it, which also suppresses the contralateral ARTR through contralateral inhibition. Next, ARTR activity decays slowly, again biasing turns to the same direction. This scheme could generate sequences of turns in the same direction. Switches in turn direction might arise from spontaneous activity in the ARTR population, from spontaneous or evoked input from neurons upstream of the ARTR, or when ARTR activity has decayed sufficiently so that it no longer exerts a bias on turning direction. To test this model, synaptic connections between ARTR clusters and between the ARTR and vSPNs need to be established, and ARTR activation and decay dynamics must be carefully characterized using electrophysiology. It should also be determined, for example with the aid of optogenetic silencing, whether the ARTR operates autonomously or whether it relies on interactions with other populations, such as cells in Rh4-6.

Comprehensive, whole-brain, cell-level imaging was crucial for discovering the ARTR circuit (*Vladimirov et al., 2014*; *Freeman et al., 2014*). Lacking a priori hypotheses regarding the location of circuits governing a behavior, the near-complete coverage of this approach helps ensure that neurons with response properties of interest, if present, will likely be identified (depending on the sensitivity of the activity reporter [*Chen et al., 2013*] and the design of computational approaches [*Freeman et al., 2014*]). Thus, while inputs to the identified neural populations certainly shape circuit activity, our measurements and analyses suggest that the identified cells are likely to be the primary set of neurons consistently involved in generating directional behavior. Of these neurons, we decided to causally interrogate the ARTR because of its strong stereotypy across fish, tight correlations to the slow switching structure of spontaneous behavior, and its pattern of projections to the premotor reticulospinal system. We identified, by ablation and stimulation, the ability of the ARTR to bias turning direction. The properties of the ARTR, including its morphological projections, activation after a turn, ability to bias turns, and slow decay time, together with the change in the temporal structure of behavior following ARTR lesions, establishes the ARTR as an important candidate for the circuit generating temporal structure in spontaneous turn sequences. However, the ARTR may also be part of a larger circuit performing this operation. The contribution of other functionally identified regions, which were mapped carefully across fish utilizing the Z-brain atlas, will be explored in future studies.

Tethered preparations are an important tool for studying the relationship between neuronal activity and behavior (*Dombeck and Reiser, 2012*). Differences between real and tethered behaviors are usually present (*Dombeck and Reiser, 2012*), but are, in many cases, small enough to allow for the study of related neural activity, including for relatively complex phenomena such as spatial representations (*Seelig and Jayaraman, 2015*; *Harvey et al., 2009*). Here, we observed differences between the fictive behavior and freely swimming behavior, such as an approximately 1.4- to 2.4-fold difference in swim frequency, but these differences were modest enough for the essential properties of

the behavior to persist, allowing the underlying signals to be analyzed. Since the length of behavioral sequences was similar when analyzed over number of swim bouts, it is possible that the time constant of the ARTR was slightly longer in the fictively behaving fish, potentially due to influences such as a lack of proprioceptive feedback. However, the persistent similarity of behavioral sequences and other behavioral kinematics (*Figure 1—figure supplement 4D,E*) between fictively and freely swimming fish suggests that the function of neural circuits underlying behavior remain largely intact under the microscope. Of course, subsequent perturbation studies, like the ones performed here, are crucial for establishing the necessity or sufficiency of neurons for a given behavior.

Based on PA-GFP tracing, neurotransmitter phenotyping, and references to the Z-Brain atlas, we have developed predictions for specific connectivity between the medial and lateral ARTR clusters, and the medial ARTR cluster and downstream reticulospinal premotor neurons. Deciphering the nature of these connections will be essential for a precise mechanistic understanding of ARTR dynamics and their link to spontaneous behavioral bias. Future experiments employing electrophysiology, viral tracing, and connectomics will verify and expand on the precise mechanistic operations of the circuits underlying spontaneous behavior in larval zebrafish.

The initiation of locomotion in other animals, such as lamprey (*Sirota et al., 2000*), salamander (*Cabelguen et al., 2003*), and cat (*Shik et al., 1969*), has been attributed to the mesencephalic locomotor region (MLR). While the MLR has yet to be identified in the larval zebrafish (*Severi et al., 2014*), the midbrain tegmental nMLF has been shown to regulate swimming (*Severi et al., 2014*; *Thiele et al., 2014*; *Wang and McLean, 2014*). Consistent with this, we find the nMLF to be routinely correlated with spontaneous swimming, and it is possible that the additional mesencephalic cells not part of the nMLF but correlated strongly with swim amplitude (*Figure 3E*, *Figure 3—figure supplement 2*) may be part of the larval zebrafish MLR. Furthermore, motor-related signals were present, albeit weakly, in other areas including in the forebrain; it will be exciting to discover whether these areas, or even the ARTR itself, are homologous to structures known to be involved in motor control in other species but have not yet been located in the larval zebrafish. That being said, while we have reported a causal role of the ARTR in spontaneous swim patterns, we do not claim that the initial command for motion originates from the ARTR. Rather, we suggest that the ARTR exerts a bias on the direction of swim bouts initiated by other circuits. This view is supported data presented in *Figure 4—figure supplement 1A*, *bottom*, which shows that additional turns are not recruited by ARTR stimulation. Thus, we argue that the ARTR occupies a position complementary to canonical motor control centers.

The neural populations uncovered by our analysis are involved in setting the direction of spontaneous swimming but may be involved in other functions as well. In principle, signals from other motor modalities as well as sensory systems could be integrated into ARTR activity fluctuations. In feature-poor environments, the ARTR system may interact with sensory systems in such a way that ARTR control of exploration is preserved but biased by the influence of weak sensory inputs. When stronger sensory cues are encountered, navigation systems purely driven by sensory stimulation may take over. For instance, all or part of the cell population may be involved in the optomotor response, as the ARTR responds to whole-field motion, and turns in the direction of motion persist after visual stimuli disappear (*Figure 5—figure supplement 3*). In the context of phototaxis, navigational strategies have been observed in larval zebrafish that also exhibit strong temporal correlations in turning direction (*Chen and Engert, 2014*), potentially involving the ARTR. Furthermore, preliminary observations show that eye movements (*Miri et al., 2011*) and turning are correlated in larval zebrafish, and activity in the vicinity of the abducens and oculomotor nuclei is correlated to turning and ARTR activity (albeit much more weakly than the strength of correlation between the ARTR and turning, *Figures 2,3*). Thus, it is possible that multiple motor patterns are represented in and coordinated by the ARTR. In the future, it will be exciting to study just how much the ARTR intersects with these complementary systems.

According to our modeling of swim trajectories generated by temporally correlated vs. uncorrelated turns, the slow fluctuations in turn direction that we observed may increase foraging efficiency under the condition that the fish are restricted to a local search (due to, for example, dangers arising from venturing into unknown places). Future work can investigate whether this strategy adapts to changes in the environment or internal state. Food restriction or low light levels, for instance, may decrease state length in order to increase diffusion and promote exploration of completely novel

environments. Conversely, favorable conditions may increase state length so as to decrease the rate of diffusion while encouraging efficient sampling of the local environment.

In summary, the whole-brain analysis, neural perturbation experiments and anatomical characterization together reveal a circuit contributing to the patterning of a spontaneous, self-generated behavior. While this circuit is likely supported by other neurons and regions, we speculate that its function may be to guide animals through environments where guidance from external cues is lacking, a context where animals must rely primarily on the internal drive of brain-autonomous activity.

## Materials and methods

All experiments presented in this study were conducted in accordance with the animal research guidelines from the National Institutes of Health and were approved by the Institutional Animal Care and Use Committee and Institutional Biosafety Committee of Janelia Research Campus. Statistical tests reported were two-tailed. Most statistical tests performed are Wilcoxon rank sum or paired signed rank (where applicable) tests because in most cases data were not normally distributed. No sample size calculations were performed, but even the experiment with the lowest sample size (N = 5 fish, sham ablation, *Figure 4B*) has statistical power over 99% for alpha = 0.05 (with z-statistics), given the large size of the ablation effect in associated experimental groups. For most summary analyses, we averaged across biological replicates, such that numerical data from each fish was weighted equally (across fish), unless indicated otherwise (across all trials, events, turns, or time – i.e. technical replicates). All error bars are mean ± SEM unless noted otherwise.

### Spontaneous swimming experiments and analysis

Larvae (5–9 dpf) were monitored in a 9.2-cm petri dish (VWR). A high-speed camera (Mikrotron 1362, Mikrotron GmbH, Germanyor AVT Pike, Allied Vision Technologies GmbH, Germany) equipped with a lens (CF35HA-1, Fujinon, Japan) running at 200 or 100 fps captured swim dynamics. Custom-written C# software (available upon request) recorded fish center of mass and orientation as fish swam spontaneously in the arena. Uniform neutral gray background illumination was delivered with a DLP projector (Dell M109S, Dell, Round Rock TX) and reflected by a 3 x 4 inch cold mirror (Edmund Optics, Barrington NJ) underneath the petri dish. The petri dish rested on a clear acrylic platform (McMaster-Carr, Elmhurst IL) equipped with a diffusive screen (Cinegel, Rosco, Stamford CT). An array of LEDs at an IR wavelength of 810 nm was used to illuminate the arena from below. An IR band pass filter (BP850, Midwest Optics, Palatine IL) allowed the IR light to reach the camera, creating an image of the fish, while blocking the visible light from the projector.

After data collection, swimming was analyzed using Matlab (Mathworks, Natick MA). Swim trajectories (fish center of mass over time) were first smoothed with a 400 ms Gaussian kernel (~40% interbout interval) with σ = 70 ms to reduce noise in recorded center of mass. Swim events were then marked at time points where instantaneous linear velocity crossed a threshold that minimized false positives and negatives. Because measurements of instantaneous velocity depend on spatial resolution and pixel noise, this threshold was adjusted for each type of recording: 3.3 mm/s for Pike camera experiments at 200 fps and 1.0 mm/s for Mikrotron camera experiments at 100 fps. For a subset of experiments (4/19 fish), fish position was recorded as the darkest point on the fish (i.e. one eye). Because this introduced an additive baseline velocity during swim events, a threshold of 4.5 mm/s was used for these experiments. Visual inspection of heading direction traces showed that each threshold yielded consistent turn classification. Turn angle was calculated as the change in heading angle during a swim bout, calculated as the difference between the heading angle 250 ms after and 250 ms before peak swim velocity.

For analyses of cumulative signed turn direction and streak length, we only considered turns that were executed at least 1 cm from the edge of the petri dish in order to eliminate artifacts arising from thigmotaxis (the propensity of fish to hug the walls of an enclosure) and avoiding artifacts from wall visibility. The cumulative sum of signed turn sequences triggered on a switch in turn direction (that is, sequences of 1 and -1, with positive values representing turns in the same direction as the switch) were then averaged over all such sequences for a given fish. To determine the last turn from a triggered switch in direction that was reliably in the same direction as the first (i.e. the average length of a turn state), we looked for where the change in turn direction within a sequence across all fish was no longer significantly different from 0 (p>=0.05, signed rank test), which corresponds to the expected value for

a fish turning left and right randomly with or without a bias. This point can also be seen as the turn (from a switch) where the average cumulative angle plateaus. Streak length was defined as the number of turns executed in the same direction before a turn in the opposite direction. Source data and analysis (*Figure 1—source data 1* e.m) are provided as supplementary files.

## Fictive behavior setup and analysis

The fictive behavior setup has been previously described in *Ahrens et al. (2012)* and the directional decoding strategy is as in *Ahrens et al. (2013)* with minor improvements. Larval zebrafish (5–7 dpf) were paralyzed by immersion in a drop of fish water with 1 mg/ml alpha-bungarotoxin (Sigma-Aldrich) and embedded in a drop of 2% low melting point agarose, after which the tail was freed by cutting away the agarose around it. Two suction pipettes – of diameter 45 μm – were placed on the tail of the fish at intersegmental boundaries, and gentle suction was applied until electrical contact with the motor neuron axons was made, usually after about 10 min. These electrodes allowed for the recording of multi-unit extracellular signals from clusters of motor neuron axons, and provided a readout of intended locomotion (*Ahrens et al., 2012*; *Masino and Fetcho, 2005*). Extracellular signals were amplified with a Molecular Devices Axon Multiclamp 700B amplifier and fed into a computer using a National Instruments data acquisition card. Custom software written in C# (Microsoft, Redmond WA) recorded the incoming signals. Fictive swim bouts were processed as described previously (*Ahrens et al., 2012*; *2013*), separately for the left and the right channels, so that the filtered signal consists of the standard deviation of the raw signal in 10 ms time bins. Subsequently, the channels were normalized by dividing the filtered signal by the average filtered signal amplitude during swim events, to account for different signal strengths that may arise from differences in the quality of the left and right recordings. Prior to averaging, each swim bout was weighted by a normalized rising exponential function, to take into account the fact that turns affect the start of swim bouts more heavily than the end of swim bouts, so that weighting the ends of swim bouts more heavily will reduce the effect of turning and lead to more robust normalization of the two channels. To determine fictive turn amplitude and distance, filtered left and right fictive signals at swim bouts were first weighed with a decaying exponential function ($\tau$ = [bout duration] / 3) to emphasize the initial bursts that determine overall turn direction. The power of the right channel was then subtracted from the power of the left channel to arrive at turn amplitude and direction, and the powers were summed to provide a measure of swim vigor or distance. We then analyzed turn history and streak length from these processed turn sequences, as outlined in *Spontaneous swimming experiments* above. For reconstructing the virtual swim trajectories, we assumed similar distributions of turn angles in the fictive and freely swimming cases, and thus converted the raw fictive turn amplitudes and directions to turn angles via normalization to an estimate of the maximum turn angle observed in freely swimming fish (150°, from the data used for *Figure 1*). Virtual distance units were defined as the square root of the summed fictive power, which approximated the distribution of bout lengths observed in freely swimming fish. Together, these virtual turn and distance units were used to calculate a sequence of virtual fish positions before and after each fictive turn bout.

## Verifying the accuracy of fictive turn direction decoding

After signals from the two electrodes, recording from peripheral motor nerves on both sides of the tail, are normalized (see *Fictive behavior setup and analysis*, above) to account for differences in signal amplitude, the power on the left and right channels are compared, exponentially weighted to emphasize the start of the fictive swim bouts, which carry the most information about turning behavior (*Mirat et al., 2013*; *Huang et al., 2013*). We verified that bilateral fictive recordings contain sufficient information for decoding turn direction using two complementary methods. First, using recordings from three points along the tail – two anterior electrodes and one posterior electrode – we verified that turns decoded from the two anterior electrodes showed an overlap between the initial burst on the anterior electrode the ipsilateral posterior electrode (*Figure 1—figure supplement 2A–F*). This is analogous to turning in freely swimming fish (*Mirat et al., 2013*), where turns are characterized by a bend to one direction at overlapping time points along the length of the tail, whereas forward swims comprise a traveling wave along the tail with a phase offset between ipsilateral anterior and posterior tail bends. Second, we performed fictive recordings in weakly paralyzed fish that were still able to move their tails and found that fictive turn direction, as decoded only from the

electrical recordings, generated a reliable classification of the direction of physical tail movement (*Figure 1—figure supplement 2G–K*; 4% discrepancy between physical and fictively decoded turn direction). We also verified that the fictive behavior did not contain struggles or Mauthner-mediated escapes (*Figure 1—figure supplement 3*). Together, these results show that the decoded fictive turn direction reliably quantifies intended turning behavior.

## Light-sheet imaging

The light-sheet imaging experiments were performed according to the paradigm previously described (*Vladimirov et al., 2014*). We used transgenic zebrafish expressing the calcium indicator GCaMP6 (*Chen et al., 2013*) under the *elavl3* promoter, which provides near-panneuronal expression, either cytosolic as *Tg(elavl3:GCaMP6f)* or *Tg(elavl3:GCaMP6s)* or nucleus-targeted as *Tg(elavl3: H2B-GCaMP6f)*. Our data set also includes one *Tg(elavl3:GCaMP5G)* fish (*Ahrens et al., 2013*). Zebrafish larvae were embedded in a custom made chamber that allowed for electrical recordings of fictive swimming from the tail and access to light-sheet excitation laser beams from the lateral and frontal direction of the fish. The lateral beam was used to scan over the majority of the brain, while the frontal beam scanned over the region between the eyes that was inaccessible to the lateral beam, thus covering most of the brain at single-cell resolution. The detection objective was moved with a piezo so that the light sheets were always in the focal plane of the objective. Using this technique, the imaging rate was about 2 brain volumes/s (1.87 ± 0.14 s), that is every cell was imaged roughly every 0.5 s. Importantly, the lateral beam rapidly switched off whenever it was located inside a circular exclusion region around the eye, so that whole-brain imaging could be performed without directly shining the laser beams into the eye. Average laser power was set to the dimmest viable average power of 44 μW (66 μW with a sweep duty cycle of 67%) (below the range considered in ref. [*Wolf et al., 2015*]), and red background illumination was provided to mimic the freely swimming light levels as well as provide a luminous environment to mask the blue laser. The red background illumination was provided to the fish by projecting homogeneous red light with a mini projector onto a screen underneath the fish (see *Vladimirov et al., 2014* for details). Each experiment lasted between 30 and 60 min and thus contained 3000–6000 whole-brain stacks.

## Analysis of volumetric light-sheet data

Imaging data were analyzed using the open-source Thunder library described in *Freeman et al. (2014)*. Thunder uses the Apache Spark cluster computing platform for manipulating and analyzing large-scale image and time series data. All analyses described here were performed on a local cluster, but can be reproduced identically on cloud compute, and sample data is made available on Amazon S3 (see below).

Using Thunder, light-sheet data were first registered by cross-correlation to a reference volume, and each voxel's time series was converted to $\Delta$F/F. We then developed a regression analysis to capture the extent to which neuronal responses were predicted by directionally specific behavior. First, two one-dimensional parameters were derived from the fictive swim signals: one capturing the instantaneous amplitude of swimming (strictly positive), and another capturing the instantaneous direction (positive for right, negative for left). We noted that, across many experiments, these two parameters tended to fall within the same region of a two-dimensional space (after normalizing amplitude to have a maximum of 1) (*Figure 2B*). To compute neuronal tuning within this space, we expanded the instantaneous value of the two signals into a nonlinear basis; intuitively, this corresponds to dividing the two-dimensional space into several small wedges each corresponding to a range of directions and amplitudes. We used a polar basis, separably and evenly tiling amplitude (three bins) and angle (four bins). Each basis had a flat top and raised cosine transition region, with 50% overlap; see *Simoncelli et al. (1992)* and *Freeman et al. (2011)* for the parameterization of this basis (*Freeman and Simoncelli, 2011*), which is more commonly used to tile the two-dimensional Fourier domain. With this basis, we represented instantaneous behavior with 12 predictor time series, each 1 x T, where T is the duration of an experiment. These predictors were each convolved with an impulse kernel k intended to reflect typical calcium dynamics; the kernel had a linear rise of 1 s and a linear decay of 5 s; variations of the kernel both in shape (e.g. exponential decay) and timing (0.5 s rise and 2 s decay) yielded qualitatively similar maps. Along with a constant offset term,

this yielded a 13 x *T* predictor matrix **X**. We then used ordinary least squares regression to infer the best fitting coefficients *b*:

$$\mathrm{b} = (\mathrm{XX}^{\mathrm{T}})^{-1}\mathrm{X}^{\mathrm{T}}\mathrm{r}$$

where *r* is the *T* x 1 fluorescence time course of either a single voxel or a neuron. The 12 coefficients (ignoring the constant) describe tuning with the two-dimensional behavioral space (e.g. polar wedge plots in *Figure 2C,F*), and $R^2$ from the regression captures prediction accuracy. Computing a weighted angular mean yields a single laterality index, used to determine hue in computational maps (*Figure 2D–F*, *Figure 3*). Note that a bilinear model (*Ahrens et al., 2008*) could have been used to estimate behavioral tuning and temporal kernel simultaneously, but preliminary analyses showed that tuning was largely invariant to the shape of the temporal kernel.

An example data set (one of the same data sets used to generate maps in *Figure 2*) is available on Amazon S3 at s3://neuro.datasets/ahrens.lab/spontaneous.turning/2/, including both neural data and behavioral regressors. And an example analysis in the form of a Jupyter notebook are included as Supplementary Files (spontaneous-turning.html, spontaneous-turning.ipynb); the notebook shows how to load data from that URI and generate a map for one of the data sets shown in *Figure 2*.

## Registering light-sheet data to the Z-Brain atlas

To register light-sheet data to the confocal data in the Z-Brain (*Randlett et al., 2015*), we used the Computational Morphometry toolkit (CMTK, https://www.nitrc.org/projects/cmtk/). To solve this cross-modal registration problem, we used two different strategies for the two types of transgenes (nuclear and cytoplasmic GCaMP). For *Tg(elavl3:GCaMP6f/s)* registrations, we used a single *Tg (elavl3:GCaMP6f;elavl3:H2B-mKate2)* fish to create a bridging reference brain from the light-sheet data to the Z-Brain. This fish was imaged on the light-sheet microscope live, then imaged again by confocal microscopy live, fixed in 4% PFA and stained with tERK, and then finally the transgene signals and tERK stain were imaged by confocal microscopy. We then used CMTK to calculate the morphing transformations through each of these steps using the *Tg(elavl3:H2B-mKate2)* to align the light sheet → live confocal → fixed confocal data, and then the tERK stain to align the to Z-Brain reference brain. To register the three *Tg(elavl3:GCaMP6f)* and one *Tg(elavl3:GCaMP6s)* fish in this study, each fish is aligned to the light-sheet volume of the bridging brain using the *Tg(elavl3: GCaMP6f)* signal, and then the 4 transformation steps (1 fish specific, 3 common to all fish) are concatenated and applied using CMTK's 'reformatx' tool.

For *Tg(elavl3:H2B-GCaMP6f)*, the anatomical volume of the 11 fish imaged in this study were all registered to a single template *Tg(elavl3:H2B-GCaMP6f)* fish. These 12 volumes were averaged, and then this mean-volume registered to the *Tg(elavl3:H2B-RFP)* (*Randlett et al., 2015*) Z-Brain volume, which is the average of 10 fish. The two transformation steps for each fish are then concatenated and applied using 'reformatx.' To confirm the accuracy of alignment, we compared the positioning of reticulospinal cells imaged live on the light-sheet microscope to the same label in the Z-Brain, which revealed good overlap (*Figure 3—figure supplement 1b*), thus validating the accuracy of our alignment in this area.

To analyze the anatomical features of the *Tg(elavl3:H2B-GCaMP6f)* derived functional volumes, we used the Z-BrainViewer and the 'ZBrainAnalysisOfMAP-Maps' function (*Randlett et al., 2015*) to compare the positioning of features with regions and cell type labels in the Z-Brain atlas (supplementary Data: SupplementaryData_ZBrainAnalysisOfNucMaps.xls).

## Identification of the ARTR in two-photon data for cell activation, ablation, and neuroanatomy

Because the location of the ARTR was stereotyped across fish (*Figure 2E*), it was possible to find the area of the ARTR using two-photon microscopy. We imaged single planes in this area, and then analyzed the imaging data rapidly in Matlab (MathWorks) as follows. First, putative ARTR cells were identified according to activity profiles; next, an ROI was drawn manually around a cluster of such identified cells, and the ΔF/F time course of this ROI was correlated, pixel-by-pixel, to the entire movie. This resulted in an image of correlation coefficients (e.g. *Figure 4H*, *Figure 5A*, *Figure 5—figure supplement 1D*) which was then verified to exhibit the structure of the ARTR. Based on this

image, cells were selected for ablation, optogenetic stimulation, photo-conversion, or overlap with dsRed and RFP (for identifying vglut2a and gad1b expressing cells, respectively).

## Optogenetic stimulation of the ARTR

First, (*Tg(elavl3:ReaChR-TagRFP-T); (Tg(elavl3:H2B-GCaMP6f)*) fish (that express both the channel-rhodopsin ReaChR (*Lin et al., 2013*) and GCaMP6f in most neurons) were embedded in agarose and imaged with a two-photon microscope while simultaneously recording fictive swimming (as described above). Regions belonging to one of the four ARTR clusters (15–20 cells in a single plane) were selected as described above. Next, selected regions were optogenetically activated using laser raster scanning (1050 nm, 50 mW) once every 80 or 100 s. Each stimulus consisted of a sequence of 7 or 12 pulses, with 200 ms pulse duration and 500 ms inter-pulse interval. The selected region was scanned at 20 Hz, that is four times per pulse, 28–48 times per stimulus.

## ARTR lesion experiments and analysis

Before ARTR lesions, fish were filmed with a high-speed camera and their behavior quantified as below. Next, we embedded the fish and used two-photon imaging to identify the ARTR according to anatomical location and function (see *Identification of the ARTR in two-photon data*, above). Next, 10 to 20 ARTR neurons (19 ± 6, mean ± SD, i.e. about 32% of the total number of neurons in the medial cluster) were selected from either the left or right medial cluster and ablated with a two-photon laser (850 nm, 120–135 mW). During the exposure, the laser beam spiraled over a circle of 1 μm diameter. To improve the specificity of ablation, laser exposure was minimized (typically 0.1–10 s, depending on e.g. the expression level of the fluorophore in the neuron) by a feedback control system using custom software: large and sudden brightness increases, which indicate successful lesions, were detected automatically and used to terminate the laser exposure. We estimated by eye that in each experiment, no more than two neurons outside of the set of targeted neurons were unintentionally damaged. For the sham ablation group, 8–10 neurons in the vicinity of the ARTR, but not in the pool of neurons activated in synchrony with ARTR neurons, were chosen randomly, and these were ablated using the same procedure described above.

All behavior experiments were performed with an AVT Pike at 30 fps. We only considered turns that were executed at least 1 cm from the edge of the petri dish in order to eliminate artifacts arising from thigmotaxis (the propensity of fish to hug the walls of an enclosure) and avoiding artifacts from wall visibility. Due to a higher noise level and lower frame rate in these Pike recordings, swim trajectory was used to calculate turn angle, and a threshold of 0.17 radian per turn was applied (without threshold change in bias: p=0.006, signed rank test). In order to compare the streak distribution of each fish before and after ablation to a random distribution of streaks, we used the normalized root-mean-square error (NRMSE) to assess goodness-of-fit. For each fish before and after ablation, we used the overall turn bias (# of left turns/# total turns) to generate a binary sequence of left and right turns equal in length to the number of recorded turns in the real data set. Individual turn identities in these sequences were determined by generating a random number between 0 and 1 and comparing its value to the overall turn bias. For instance, if the overall bias was 0.4, any random number less than or equal to 0.4 was called a left turn, and any random number greater than 0.4 was called a right turn. For our analyses, the NRMSE was then calculated comparing the streak histograms for random, simulated sequences to the streak histograms for real fish (which, pre-ablation, show strong, non-geometric correlations), using:

$RMSE = \sqrt{\frac{\sum_{i=1}^{15}(obs_i - coin_i)^2}{15}}$, where $obs_i$ and $coin_i$ represent the relative frequency of the $i^{th}$ streak length for the observed distribution and associated 'coin flip' distribution, respectively, and $NRMSE = \frac{RMSE}{obs_{max} - obs_{min}}$, where $obs_{max}$ and $obs_{min}$ represent the maximum and minimum relative frequency for the observed distribution, respectively.

As two distributions increase in similarity, the NRMSE approaches 0.

## Identification of neurotransmitter phenotype

*Tg(vGlut2a:dsRed);Tg(elavl3:H2B-GCaMP6f)* and *Tg(gad1b:RFP);Tg(elavl3:H2B-GCaMP6f)* fish at 6dpf were embedded in 2% agarose in a 35 mm petri dish. The fish were imaged under a two-photon microscope at 930 nm in the anterior hindbrain at the level of rhombomeres 2–3. Several planes

about 2 µm apart were imaged in order to find a plane where all four clusters of the ARTR were strongly visible. The ARTR was functionally identified by using a correlational measure (see *Identification of the ARTR in two-photon data*, above) to construct a correlational map over the various planes. The plane best depicting all four clusters was selected and imaged in the red channel (1005 nm). The images were then superimposed to evaluate the expression pattern of vGlut2a:DsRed or gad1b:RFP in ARTR neurons.

## Neurite-tracing experiments

*Tg(alpha-tubulin:C3PA-GFP);Tg(elavl3:H2B-GCaMP6f)* or *Tg(alpha-tubulin:C3PA-GFP);Tg(elavl3:jRCaMP1a)* fish at 6dpf were embedded in 2% agarose in a 35-mm petri dish. For some experiments, the reticulospinal neurons were retrogradely labeled with 20% alexa-680-dextran according to published protocols (*Fetcho and O'Malley, 1995*). The fish were imaged under a two-photon microscope at 930 nm (1050 nm for jRCaMP1a) in the anterior hindbrain at the level of rhombomeres 2–3. The ARTR was functionally identified by using a correlational measure to construct a correlation map as described above. Individual cells of either the medial or lateral cluster were selected on one side of the brain in a plane containing sections of all four clusters. We modified the neurite tracing protocol developed by *Datta et al. (2008)* (*Datta et al., 2008*) to trace projections from a subset of ARTR neurons. Cells were selected using custom written software and PA-GFP was activated using a protocol for iterative activation: ten 250 ms pulses of 780 nm pulsed infrared laser light were administered over a course of 16 cycles spaced 15 min apart for 4 hr. Selective activation was confirmed after each cycle by switching to 930 nm and imaging the selected plane for increased fluorescence. At the end of 4 hr, fish were transferred to an incubator and kept in the dark for another hour to allow additional time for GFP transport along the neurites. Subsequently, the fish were imaged on a Zeiss 710 confocal microscope using a 20x or 40x objective. The confocal stacks were then analyzed using the Simple Neurite Tracer plugin in FIJI (ImageJ).

## Transgenic zebrafish

Transgenic zebrafish larvae used in this study were in either *casper* or *nacre* background (*White et al., 2008*). TgBAC(gad1b:loxP-RFP-loxP-GFP), described previously (*Satou et al., 2013*) and TgBAC(slc17a6b:loxP-DsRed-loxP-GFP), described previously (*Satou et al., 2013*; *Koyama et al., 2011*) were used in the absence of Cre mediated recombination and are referred to as *Tg(gad1b:RFP)* and *Tg(vGlut2a:dsRed)*, respectively. *Tg(alpha-tubulin:C3PA-GFP)* was used as described previously (*Ahrens et al., 2012*). The *Tg(elavl3:GCaMP6f)*[jf1], *Tg(elavl3:H2B-GCaMP6f)*[jf7] (*Freeman et al., 2014*; *Chen et al., 2013*; *Kanda et al., 1998*; *Quirin et al., 2016*), and *Tg(elavl3:ReaChR-TagRFP-T)*[jf10] (*Lin et al., 2013*) and *Tg(elavl3:jRCaMP1a)*[jf16] and *Tg(elavl3:H2B-mKate2)*[jf14] lines were newly generated using the Tol2 system (*Urasaki et al., 2006*) and a published *elavl3* sequence (*Sato et al., 2006*). The fish lines are being deposited to ZIRC and the DNA constructs to Addgene, and are also directly available from Janelia Research Campus upon request. The larvae were reared at 14:10 light-dark cycles according to the standard protocol at 28.5° C (undefined).

## Markov model for fitting and generating turn sequences

For each fish, we trained a hidden Markov model with the signed sequence of all valid turns (i.e. at least 1 cm from dish edge) using a forward-backward Baum-Welch algorithm (hmmtrain, Matlab) to form a Markov model of the experimentally observed turn sequences. This algorithm terminated when the change in the log likelihood that turn sequences were generated from estimated transition and emission probabilities, the change in the norm of the transition matrix, and the change in the norm of the emission matrix were all less than $10^{-6}$, or after 2000 iterations (4 of 19 fish, all metrics less than the default tolerance of $10^{-4}$) This method produced best-fit estimates for the underlying transmission and emission probabilities explaining the turn sequences for each fish. We then generated sequences of binary turns (hmmgenerate, Matlab) equal in number to the turns used to train the model for each fish. These simulated turn sequences were then used to analyze turn history and streak length, as outlined above.

For simulating exploration (*Figure 6*), two initial models, a correlated fish and a 'random' fish, were used to generate $10^6$ swim trajectories of 40 bouts each. For the correlated fish, emission sequences were generated with $P_{transition} = [P_{L \blacktriangleright L}\ P_{L \blacktriangleright R}\ P_{R \blacktriangleright L}\ P_{R \blacktriangleright R}] = [0.86\ 0.14\ 0.15\ 0.85]$ and

$P_{emission}$ = [$P_{turn\ L\ |\ L}$ $P_{turn\ R\ |\ L}$ $P_{turn\ L\ |\ R}$ $P_{turn\ R\ |\ R}$] = [0.85 0.15 0.10 0.90], the best-fit probability matrices for the fish in *Figure 1B*. For the 'random' fish, emission sequences were generated with $P_{transition}$ = [0.5 0.5 0.5 0.5] and $P_{emission}$ = [1 0 0 1]. These Markov sequences were then used to assign direction to individual turn magnitudes, which were drawn according to the turn angle probability distribution derived from all acquired swim bouts in freely swimming fish.

The 'random' model fish diffuses on average more rapidly away from the starting point due to the reduced 'winding' properties of the trajectories. Assuming pressure to not venture too far afield due to potential dangers in faraway areas, we matched the average diffusion rate of the correlated and the random model fish after 40 swim bouts. In one random model, diffusion was matched to the correlated fish by decreasing simulated bout length by 24.6%. In the other model, diffusion was matched by broadening the turn angle probability distribution, resulting in a 47.0% increase in mean turn angle. The $10^6$ trajectories for each model, which each started at a common point in space but with a random initial heading direction, were then used to measure how many virtual "resources" were collected by each fish model. Virtual resources were distributed randomly over an area approximately 25 bout lengths x 25 bout lengths in size with density 0.10 resources/bout length. Resources were counted as collected if simulated trajectories passed within 3.25 bout lengths of a resource position, representing a remote detection radius such as an odor gradient. We then used resource collection as function of bout number or angle turned – each of which is energetically and temporally costly – to assess model exploration efficiency.

Although these simulations suggest significant differences in foraging efficiency between correlated and 'random' swim strategies over a large number of fish or a long period of time (i.e. $10^6$ simulated trajectories), we sought to characterize the distribution of our efficiency statistics. We used this characterization to inform hypothesis tests and provide an estimate of the number of individuals (or length of time) required to find significant increases in efficiency for the correlated model. For the SEM indicated by error bars in the right panels of *Figure 6D* and *Figure 6F*, we determined critical simulation number (N) by examining graphs of gathered resources after 40 swims over increasing numbers of simulated trajectories (left panels of *Figure 6—figure supplement 1E and G*, respectively). Specifically, we defined N to be the number of simulated trajectories needed for this quantity (gathered resources) to stabilize (*Koehler et al., 2009*), and we considered the quantity to be stable when the local Fano factor (variance/mean over a moving window of 400 simulations) first dropped below $10^{-5}$ for either the correlated model or each respective 'random' model. This threshold, an automatic asymptotic indicator, consistently aligned with visual inspection of the graphs. We then used N (1304 for the bout length normalized and 1520 for the turn angle normalized 'random' model comparisons) for hypothesis testing and for determining mean ± SEM in plots of resource/swim. Because the distributions of angle/resource depend on the underlying 'success' rate at each resource bin, that is almost all simulated trajectories encounter at least 1 resource and only a small fraction of simulated trajectories encounter 10 resources after 40 swims, we determined N separately across 'successful' simulations for each bin when plotting angle/resource (left panels of *Figure 6D* and *Figure 6F*). For comparisons with the bout length normalized model, N = 2340 at 1 resource (with 91.0% 'success' rate for 'random', n = 2129 trajectories; with 95.3% 'success' rate for correlated, n = 2191 trajectories) and N = 34,810 at 10 resources (with 0.20% 'success' rate for 'random', n = 68 trajectories; with 1.02% 'success' rate for correlated, n = 354 trajectories). For comparisons with the turn angle normalized model, N = 2340 at 1 resource (with 93.6% 'success' rate for 'random', n = 2191 trajectories; with 95.3% 'success' rate for correlated, n = 2231 trajectories) and N = 33,260 at 10 resources (with 0.81% 'success' rate for 'random', n = 270 trajectories; with 1.02% 'success' rate for correlated, n = 338 trajectories).

In a complementary analysis, we also plotted the two-tailed Student's t-test p-value and statistical power for efficiency statistics over number of simulated trajectories. These analyses, which depend on the underlying mean and variance of each quantity, reveal how many individuals need to exist (or swim bouts need to occur) before the underlying advantage of the correlated model can be realized consistently.

## Supplementary methods

### Electrical stimulation of the ARTR

Stimulation pipettes were pulled from theta glass capillaries (~10 MΩ, tip diameter ~2 µm), and then coated with nano-gold, which made the pipette visible with 930 nm laser illumination (unpublished data). Zebrafish larvae (5–7 dpf; *Tg(elavl3:GCaMP6f)*) were paralyzed with a-bungarotoxin, embedded in 2% low melting point agarose, and immersed in external solution (in mM: 134 NaCl, 2.9 KCl, 2.1 $CaCl_2$, 1.2 $MgCl_2$, 10 HEPES, and 10 glucose, pH = 7.8). A small piece of skin above the hindbrain was cut open for pipette insertion. Fictive behavior was recorded using two suction pipettes as described in *Fictive behavior setup*. A stimulation pipette was inserted diagonally through the cut in the skin and targeted to the lateral edge of the medial cluster of the ARTR, which was monitored with a two-photon microscope. Next, the pipette was advanced further into the ARTR, and test electrical pulses were delivered while ARTR activation was monitored with the two-photon microscope. The pipette was moved in the dorsal-ventral direction, rostral-caudal direction and further towards the midline until a point was reached where maximal ARTR activation to the test pulse was observed. The distance from the initial position at the edge of the ARTR was typically in the range of 20 µm and maximally 40 µm (note that the dorsal-ventral extent of the ARTR is about 80 µm). A brief electrical shock train (duration: 0.2–2 ms, inter-pulse-interval: 20–50 ms, number of pulses: 5–10, voltage 10–30 V) was delivered through the pipette to activate the ARTR neurons.

### GABA immunohistochemistry for additional validation of neurotransmitter phenotype determination

In addition to the experiments of *Figure 5A,B* we verified the neurotransmitter phenotype of the lateral ARTR clusters using immunohistochemistry. *Tg(α-tubulin:C3PA-GFP);Tg(elavl3:H2B-GCaMP6f)* fish at 6dpf were embedded in 2% agarose in a 35 mm petri dish. The fish were imaged under a two-photon microscope at 930 nm in the ARTR region. Several planes about 2 µm apart were imaged in order to find a plane where all four clusters of the ARTR were visible. The ARTR was functionally identified as in *Figures 4*, *5*. Individual cells of either the medial or lateral cluster were selected on one side of the brain. Cells were selected using custom written software and PA-GFP was activated at 780 nm (250 ms pulses, 10 cycles, 15 min apart for 2 hr). The fish were then released from the agarose, euthanized with an overdose of MS-222 (tricaine) and transferred to a dish containing 10mM EGTA in PBS for 1 hr. The fish were then fixed overnight in 4% PFA-PBS pH 7.4 at 4°C. Fixed fish were then dissected to expose the brain and processed for immunohistochemistry using published protocols (*McLean and Fetcho, 2004*). A primary antibody to GABA (1:400; Sigma A2052, St Louis, MO) was used to label GABA in neurons. At the end of the procedure, fish were mounted in Vectashield (Vector Laboratories, Burlingame CA) and imaged on a Zeiss 710 confocal microscope using a 20x or 40x objective.

## Acknowledgements

We thank the Janelia GENIE project for providing reagents before publication (jRCaMP1a [*Dana et al., 2016*]); Minoru Koyama and Avinash Pujala for sharing data used in *Figure 1—figure supplement 3*; Mladen Barbic and Janelia Instrument Design and Fabrication for providing nano-gold-coated pipettes for the stimulation experiments; David Hildebrand for sharing a elavl3: GCaMP6f DNA construct; Minoru Koyama for help with anatomy experiments; Philipp Keller for help with microscopy; Emre Aksay, Alexandro Ramirez, Martin Haesemeyer, Robert Johnson, and Brett Mensh for discussions; Charles Zuker, Minoru Koyama, Karel Svoboda, Vivek Jayaraman, Andrew Bolton and Scott Sternson for reading the manuscript and providing feedback; NIH grants DP1NS082121 and R01DA030304 and HHMI for support.

## Additional information

### Funding

| Funder | Author |
| --- | --- |
| Howard Hughes Medical Insti- | Yu Mu |

| | |
|---|---|
| tute | Sujatha Narayan<br>Chao-Tsung Yang<br>Jeremy Freeman<br>Misha B Ahrens |
| National Institutes of Health | Timothy W Dunn<br>Owen Randlett<br>Eva A Naumann<br>Alexander F Schier<br>Florian Engert |
| Marie Curie Fellowship | Eva A Naumann |
| National Science Foundation | Timothy W Dunn |

The funders had no role in study design, data collection and interpretation, or the decision to submit the work for publication.

## Author contributions
TWD, Conception and design, Acquisition of data, Analysis and interpretation of data, Writing the article; YM, Acquisition and analysis of anatomical data, Contribution to writing; SN, Analysis and interpretation of anatomical data, Contribution to writing; OR, AFS, Conception and design, Writing the article; EAN, Generation of DNA constructs and transgenic zebrafish, Contribution to writing; C-TY, Analysis and interpretation of data, Contribution to writing; JF, Conception and design, Contribution to writing; FE, Conception and design, Analysis and interpretation of data, Writing the article; MBA, Conception and design, Acquisition of data, Analysis and interpretation of data, Contribution to writing

## Author ORCIDs
Owen Randlett, http://orcid.org/0000-0003-0181-5239
Eva A Naumann, http://orcid.org/0000-0002-1671-4636
Alexander F Schier, http://orcid.org/0000-0002-5317-494X
Florian Engert, http://orcid.org/0000-0001-8169-2990
Misha B Ahrens, http://orcid.org/0000-0002-3457-4462

## Ethics
Animal experimentation: All experiments presented in this study were conducted in accordance with the animal research guidelines from the National Institutes of Health and were approved by the Institutional Animal Care and Use Committee (#13-98) and Institutional Biosafety Committee of Janelia Research Campus.

## Additional files
### Supplementary files
• Supplementary file 1. Example whole-brain data and analysis code (the data is on Amazon S3; the example analysis code links to the data).

• Supplementary file 2. Table of brain areas with functional signals for the Z-Brain atlas.

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
