## [Decision Letter]

Thank you for submitting your work entitled "A hindbrain control system for exploratory locomotion" for consideration by *eLife*. Your article has been favorably evaluated by three reviewers, including Mark Masino and Ronald Calabrese, who is a member of our Board of Reviewing Editors. The evaluation was overseen by Eve Marder as Senior editor.

The reviewers have discussed the reviews with one another and the Reviewing Editor has drafted this decision to help you prepare a revised submission.

Summary:

This research report presents a technological tour de force, in which the authors use light-sheet laser imaging techniques to identify neurons in the larval zebra fish brain that bias turning direction. The authors present convincing evidence that two bilateral clusters of neurons in the anterior rhombencephalon (ARTR) are asymmetrically active during runs of same direction turns. In a uniformly lit clueless arena, free-swimming larvae tend to make runs of turns in the same direction and switch stochastically to runs of turns in the opposite direction. Such turning bias can be observed in fictive swim preparations. Ipsilateral neuron clusters in the ARTR are coactive with runs of turns to the same side. One of the clusters is shown to be GABAergic and the other glutamatergic. The GABAergic cluster projects contralaterally and is hypothesized to set up mutual inhibition between bilateral clusters. A minimal network model based on positive feedback to the ipsilateral cluster when a turn is made that decays slowly and mutual inhibition across the midline can reproduce the observed switching turn bias. A two state Markov model based on this minimal model can reproduce behavioral tracks.

The data are extensive and appropriately analyzed with relevant statistics. The Figures are easy to assimilate and the legends clear. Supplementary data answers for almost all controls. Materials and methods is extensive and complete.

Despite the demonstrable strengths of the manuscript, at present it claims to do much more than it actually achieves. Therefore, we feel that the present manuscript should be revised to describe clearly what it has accomplished, and that the authors should consider doing the considerable follow-up experiments and submitting them either as a Research Advance to this paper, assuming this one is successfully revised, or as an additional stand-alone paper, as described below.

Essential revisions:

There are two classes of concerns that make this paper more suitable as a proof of principle that the methods can reveal putative behaviorally relevant neurons in free moving fish and behaviorally relevant preparations.

1) How the behavior studied is controlled is not well discussed despite the modeling.

A) It remains unclear how the activity of the ARTR is initiated and terminated and how the switch of the activity between the left and right sides is mediated.

B) The authors show that ARTRs project to some reticulospinal neurons, however it is still unclear if these neurons are the ones relaying the turning commands to the spinal cord. Ablation or optogenetic activation of these neurons should be used to test this.

2) The claim that the ARTR neuron clusters direct biased turning is based on correlation; neither necessity nor sufficiency of the ARTR neurons is shown for directing turning bias.

A) We are not given an estimate of the number of neurons in the ARTR clusters so the results of ablations/activations are hard to evaluate critically. The effect of ablations on turning bias are small; this concern pertains both to the turn bias (Figure 4) and randomization of turn sequences (Figure 4). It is hard to see the ARTR cells as necessary for turn bias with such small effects.

Inexplicably, only the medial cluster was ablated (why?) when the inhibitory neurons of the lateral cluster should have the bigger effect. There should also be a bigger effect for more cells ablated and the effect of number of cells ablated on outcome was not evaluated.

B) The optogenetic stimulation experiments are not totally convincing. 15-20 cells in a cluster are stimulated but we are not told what portion of the cluster population this represents. Moreover, the effects are not inconsistent with simple motor commands. Only Figure 5—figure supplement 2 differentiates these cells (both medial and lateral clusters presumably?) from more 'downstream' motor commands (vSPN cells) and the time course effect is not large. The Ca responses in the ARTR cells are slower, but is this truly indicative of activity or Ca dynamics, and is it large enough to differentiate these cell types functionally? (We are not even told if the ARTR cells imaged are medial or lateral cluster cells.). The explanation in Discussion for why the stimulation experiments affect turn amplitude is not convincing.

C) A more complete set of ablation and activation experiments is needed to convincingly indicate necessity and sufficiency of the ARTR cells for directing biased turning. While these are beyond the scope of this submission, these experiments could constitute a nice Research Advance paper in *eLife* in 6 months or a year's time.

i) Specific ablations of medial and lateral ARTR cells (glutamatergic vs. GABAergic neurons) with assessment of the% of the population deleted, both unilaterally and bilaterally. How do bilateral lesions affect free swimming behavior?

ii) The authors show that ARTR project to some reticulospinal neurons, however it is still unclear if these neurons are the ones relaying the turning commands to the spinal cord. Ablation or optogenetic activation of these neurons should be used to test this.

iii) There are other regions that are equally active during the swimming activity such as Rh4-6 of a region in the caudal brainstem (caudal to IO, Figure 3). The authors do not consider that these regions contribute as much as ARTR to the behavioral pattern. However, there is no experimental support of this assumption. Ablation and stimulation of these regions should be tested to exclude they potential contribution to determining the swimming direction.

There is a more minor technical concern in addition. We are not convinced that the difference in the amplitude of the fictive activity (Figure 1) can be used as a proxy for turns. The first burst on the side of the turn is always larger and this will affect the fit and the power of the activity. The authors should analyze the duration of each burst that should be larger on the side of turn compared to the contralateral side.

Recommendation:

The authors should refocus the paper in light of the concerns above. Present the work as a major step forward in the identification of neurons involved in a complex behavior without declaring that the ARTR neurons are the main players in the control of swim turn bias. Rather the experiments to date (Ablation and stimulation) point to these neurons as important candidates that at least contribute control. In this regard the authors may want to eliminate the modeling, holding it back for a future Research Advance paper. We are taking this position because of *eLife*'s general philosophy to not require many months or extensive new experiments for a paper. The reviewers felt that this paper, in principle, deserves publication, but only if it is revised to more correctly represent what it has actually demonstrated, which is considerably less than claimed.

---

## [Author Response]

*Despite the demonstrable strengths of the manuscript, at present it claims to do much more than it actually achieves. Therefore, we feel that the present manuscript should be revised to describe clearly what it has accomplished, and that the authors should consider doing the considerable follow-up experiments and submitting them either as a Research Advance to this paper, assuming this one is successfully revised, or as an additional stand-alone paper, as described below.*

We thank the reviewers and editors for their careful reading of our manuscript and their constructive suggestions for improvement. We have assiduously considered all feedback and believe that we have revised the manuscript to address the major concerns as reiterated in the “recommendation” section below. We agree that our submission contained both a characterization of neural populations involved in driving turning behavior, and an interpretative part, embodied by a semi-mechanistic model, that was more speculative. The editor and reviewers suggested to focus on the former part and phrase the results more as a demonstration of the capabilities of the experimental and analytical system in uncovering populations involved in turning. It was also suggested that we spend more time on testing the validity of the model for a follow-up publication, which we plan to do. We have therefore removed the reduced network model from the manuscript, and recast the narrative and discussion concerning the relationship between the ARTR and temporal correlations in turn sequences as a hypothesis that still needs to be tested.

*Essential revisions:*

*There are two classes of concerns that make this paper more suitable as a proof of principle that the methods can reveal putative behaviorally relevant neurons in free moving fish and behaviorally relevant preparations.*

*1) How the behavior studied is controlled is not well discussed despite the modeling.*

*A) It remains unclear how the activity of the ARTR is initiated and terminated and how the switch of the activity between the left and right sides is mediated.*

We have added a paragraph in the Discussion (third paragraph), marked as a hypothesis and not a result, about how the switches in turning direction might come about (spontaneous ARTR activity; spontaneous or stimulus- driven input; or decay in ARTR activity removing the bias).

*B) The authors show that ARTRs project to some reticulospinal neurons, however it is still unclear if these neurons are the ones relaying the turning commands to the spinal cord. Ablation or optogenetic activation of these neurons should be used to test this.*

To address the fact that we have only shown projections of the ARTR to vSPNs but present no proof of this connection, we have removed the semi-mechanistic model that relies on this connection. As suggested by the “Recommendation,” we leave vSPN ablation and stimulation experiments for a future follow-up study.

*2) The claim that the ARTR neuron clusters direct biased turning is based on correlation; neither necessity nor sufficiency of the ARTR neurons is shown for directing turning bias.*

*A) We are not given an estimate of the number of neurons in the ARTR clusters so the results of ablations/activations are hard to evaluate critically. The effect of ablations on turning bias are small; this concern pertains both to the turn bias (Figure 4) and randomization of turn sequences (Figure 4). It is hard to see the ARTR cells as necessary for turn bias with such small effects.*

*Inexplicably, only the medial cluster was ablated (why?) when the inhibitory neurons of the lateral cluster should have the bigger effect. There should also be a bigger effect for more cells ablated and the effect of number of cells ablated on outcome was not evaluated.*

We have added the information about the total numbers of neurons in the ARTR populations (Results, subsection “Whole-brain maps reveal neural representations of spontaneous behavior” first paragraph), and apologize for not reporting this earlier. When taking this number into account, the effect of the ablation and stimulation experiments is actually not so small. When about 32% of medial-cluster cells are ablated (about 19 out of about 60), turning shifts to the opposite direction by about 18%, and when similar numbers of cells are stimulated (an area with 15-20 cells), turning increases by about 29%. It also appears we caused confusion about claiming sufficiency and necessity – we phrased it as “suggesting” sufficiency and necessity, but agree that using these words is confusing. We now simply say that the ARTR is able to bias turn direction, which we believe is strongly supported by the ablation and stimulation experiments. We are not sure we understand why ablation of the lateral cluster should have a bigger effect than ablation of the medial cluster, but we agree this is an important experiment to do. As per the “Recommendation”, we will perform this experiment, as well as more detailed quantification of size effects, as part of a follow-up study (Results, subsection “The ARTR biases spontaneous turning”).

*B) The optogenetic stimulation experiments are not totally convincing. 15-20 cells in a cluster are stimulated but we are not told what portion of the cluster population this represents. Moreover, the effects are not inconsistent with simple motor commands. Only Figure 5—figure supplement 2 differentiates these cells (both medial and lateral clusters presumably?) from more 'downstream' motor commands (vSPN cells) and the time course effect is not large. The Ca responses in the ARTR cells are slower, but is this truly indicative of activity or Ca dynamics, and is it large enough to differentiate these cell types functionally? (We are not even told if the ARTR cells imaged are medial or lateral cluster cells.). The explanation in Discussion for why the stimulation experiments affect turn amplitude is not convincing.*

As mentioned above, we now report on the total number of neurons in the ARTR clusters (Results, subsection “Whole-brain maps reveal neural representations of spontaneous behavior”, first paragraph). Regarding the temporal dynamics, thank you for this observation, it is true that potential differences in calcium dynamics confound the interpretation of the dynamics of the fluorescence signal. We have adjusted the content of the paragraph in question to acknowledge this, by emphasizing that this is an observation about calcium fluorescence, and that electrophysiology or voltage imaging is necessary to observe the temporal dynamics at the spiking level. Regarding the explanation in the Discussion, we have now removed this, as well as the related paragraph that followed. As we are no longer making claims about the precise function of the ARTR, we believe that trying to explain the effect of activation on turn angle is no longer important. In Figure 3—figure supplement 3 and Figure 5—figure supplement 2, we are reporting the pooled activity of both medial and lateral ARTR clusters, and we have now made this clear. We think that pooling is valid for these comparisons, as medial and lateral ARTR dynamics are very similar; we now show medial and lateral dynamics separately in Figure 2—figure supplement 3.

*C) A more complete set of ablation and activation experiments is needed to convincingly indicate necessity and sufficiency of the ARTR cells for directing biased turning. While these are beyond the scope of this submission, these experiments could constitute a nice Research Advance paper in eLife in 6 months or a year's time.*

*i) Specific ablations of medial and lateral ARTR cells (glutamatergic vs. GABAergic neurons) with assessment of the% of the population deleted, both unilaterally and bilaterally. How do bilateral lesions affect free swimming behavior?*

*ii) The authors show that ARTR project to some reticulospinal neurons, however it is still unclear if these neurons are the ones relaying the turning commands to the spinal cord. Ablation or optogenetic activation of these neurons should be used to test this.*

*iii) There are other regions that are equally active during the swimming activity such as Rh4-6 of a region in the caudal brainstem (caudal to IO, Figure 3). The authors do not consider that these regions contribute as much as ARTR to the behavioral pattern. However, there is no experimental support of this assumption. Ablation and stimulation of these regions should be tested to exclude they potential contribution to determining the swimming direction.*

We agree that these will be interesting and informative experiments. As suggested here and in the “Recommendation,” we leave these for the future, and have re-phrased the manuscript to better reflect the experimental results. In the revised manuscript, we have removed any suggestions that the ARTR might be sufficient and necessary for the behavior.

*There is a more minor technical concern in addition. We are not convinced that the difference in the amplitude of the fictive activity (Figure 1) can be used as a proxy for turns. The first burst on the side of the turn is always larger and this will affect the fit and the power of the activity. The authors should analyze the duration of each burst that should be larger on the side of turn compared to the contralateral side.*

We are not entirely sure we understand this comment, because we don’t use the difference in amplitude of the fictive activity, but rather the difference of a weighted integral of the power of the fictive signal (Materials and Method, subsection “Fictive behavior setup and analysis”, paragraph one). This measure does contain information about the amplitude of the fictive activity, but it also contains information about the duration of the bursts, especially of the first burst due to the exponential weighing filter. We found this to be the most robust measure of turn direction, based on three separate sets of experiments/analyses, one of which was based on the duration of the first burst.

Analysis 1: This was performed in Ahrens et al., Frontiers in Neural Circuits, 2013. Here, fictively behaving zebra fish were stimulated with left- or rightward moving gratings, which robustly elicit left and right turns, respectively, in freely swimming and head-embedded fish. Thus we were able to interpret the fictive as directional turns with high reliability. We applied 9 different decoding strategies and checked which ones separated behavioral events elicited by left- and rightward moving gratings the best. We found that a weighted difference of summed power was the most reliable (decoder #7 in panel A of Figure 2 of the Frontiers paper). On the other hand, the relative difference in the width of the first burst (decoder #1) was less reliable, most likely because this measure is less robust to noise in the data. Of course we did observe that the width of the first burst on the side of the turn was wider, but measurement of this quantity was noisier than that used by decoder #7

Analysis 2: In the current work, in Figure 1—figure supplement 2 panels A-F, we provide another verification of the decoding strategy. Very briefly, we show that a decoded turn (decoded by our strategy) is accompanied by simultaneous motor nerve signals on the side of the turn at widely separated parts on the tail, consistent with a real turn, but a decoded forward swim has phase-shifted signals at these points, consistent with a real forward swim.

Analysis 3: In Figure 1—figure supplement 2 panels G-K, we used weakly paralyzed fish to match fictive turn direction, decoded from electrical signals (that were still possible because there was only minimal movement in the tail), to the actual tail movement (that was still measurable because the tail still moved). We found strong agreement between the decoded turn direction and the physical tail movement.

If we misunderstood the comment, we will be happy to perform another analysis of the data. However, we hope that these three independent verifications provide sufficient evidence that our turn decoding strategy is accurate.

*Recommendation:*

The authors should refocus the paper in light of the concerns above. Present the work as a major step forward in the identification of neurons involved in a complex behavior without declaring that the ARTR neurons are the main players in the control of swim turn bias. Rather the experiments to date (Ablation and stimulation) point to these neurons as important candidates that at least contribute control. In this regard the authors may want to eliminate the modeling, holding it back for a future Research Advance paper. We are taking this position because of eLife's general philosophy to not require many months or extensive new experiments for a paper. The reviewers felt that this paper, in principle, deserves publication, but only if it is revised to more correctly represent what it has actually demonstrated, which is considerably less than claimed.

We agree with this recommendation, and have addressed the points raised throughout the manuscript.

1) Instead of ascribing the temporal correlations in turn direction to the ARTR, we now suggest that the ARTR is part of a network inducing the temporal correlations (e.g. Abstract; Introduction; Discussion, first, third, fourth and last paragraphs).

2) To better reflect the new focus of the manuscript, we changed the title to “Mapping the brain for neural activity controlling exploratory locomotion”.

3) Instead of suggesting that the ARTR is sufficient and necessary for biasing turn direction, we now only say that it is able to induce turn bias (as shown by the stimulation and ablation experiments; Results subsection “The ARTR biases spontaneous turning”).

4) We removed the reduced network model figure from the manuscript. We describe the model on a conceptual level in a paragraph in the Discussion (third paragraph), but make it clear that this represents a hypothesis that still needs to be tested. As suggested by the review, we hope to submit the reduced network model, together with more solid evidence, in the future as a Research Advance manuscript.

5) We re-phrased the results about the temporal dynamics of the ARTR compared to reticulospinal neurons as merely an observation about the calcium fluorescence signal, and emphasized that the real difference in time course needs to be established using electrophysiology (Results, subsection “ARTR neurotransmitter identity and morphology”, fourth paragraph).

6) We retained the emphasis on the interpretation of the morphology results as merely suggestive of a connectivity pattern that needs to be verified in the future using more targeted techniques (Results, subsection “ARTR neurotransmitter identity and morphology“, third paragraph).

7) A number of more minor changes have been made throughout the manuscript and figures to reflect the recommendation and improve the clarity and accuracy of the manuscript.